# The importance of regional sea-ice variability for the coastal climate and near-surface temperature gradients in Northeast Greenland

Sonika Shahi[1,2], Jakob Abermann[1,2,3], Tiago Silva[1,2], Kirsty Langley[3], Signe Hillerup Larsen[4], Mikhail Mastepanov[5], Wolfgang Schöner[1,2]

[1]Department of Geography and Regional Science, University of Graz, Graz, Austria
[2]Austrian Polar Research Institute, Vienna, Austria
[3]Asiaq-Greenland Survey, Nuuk, Greenland
[4]Geological Survey of Denmark and Greenland, Copenhagen, Denmark
[5]Aarhus University, Aarhus, Denmark

*Correspondence to*: Sonika Shahi (sonika.shahi@uni-graz.at)

**Abstract.** The climate in Northeast Greenland is shaped by complex topography and interaction with the cryosphere. Since the regional ecosystem processes are sensitive to atmospheric stability conditions, it is crucial to capture this complexity including adequate cryosphere coupling. This study uses an observational dataset from the Zackenberg region (Northeast Greenland) to investigate the local and large-scale factors that determine the slope temperature gradient (STG) i.e., the temperature gradient along the mountain slope. A synthesis of automated weather stations, reanalysis, and a regional climate model simulations was used. For all seasons, our results show that snow cover and near-fjord ice conditions are the dominating factors governing the temporal evolution of the STG in the Zackenberg region. Considering large-scale drivers of STG, we find that temperature inversions are associated with positive 500 hPa geopotential height and surface pressure anomalies over East Greenland. A strong connection between fractional sea-ice cover (SIF) in the Greenland Sea and the terrestrial climate of the Zackenberg region is found. A positive SIF anomaly coincides with a shallow STG, i.e., more positive (inversions) or less negative than the mean STG, since the temperature at the bottom of the valley decreases more than at the top. For example, the mean STG varies by ~4 ºC km$^{-1}$ for a corresponding ~27 % change in SIF. Reduction in temperature and precipitation (snowfall) during the days with high sea ice also affects the surface mass balance (SMB) of nearby glaciers and ice caps as shown for the A. P. Olsen Ice Cap. During summer, days with high SIF are associated with a positive SMB anomaly in the ablation area (~16 mm w.e. day$^{-1}$; indicating less melt) and a negative anomaly in the accumulation area (~-0.3 mm w.e. day$^{-1}$; indicating less accumulation). Based on our findings, we speculate that the local conditions in the Zackenberg region associated with anomalously low sea ice (i.e., a decrease in atmospheric stability) will be more prominent in the future with climate warming.

## 1    Introduction

Near-surface air temperature is a key indicator of the Earth's climate system and a powerful proxy of the surface energy balance (Ohmura, 2019). Consequently, it is widely used to describe climate change's effects on a variety of environmental

processes, particularly regarding the water and energy exchange on a local, regional, and global scale. Additionally, to show changes and sensitivity of various ecosystem components with respect to climate change, air temperature is the key variable. Knowing the spatial distribution of temperature in complex terrain, especially the relationship between temperature and elevation (i.e., lapse rate), is imperative for accurate melt modelling in glacierized or non-glacierized catchments (Chutko and Lamoureux, 2009; Hulth et al., 2010; Mernild and Liston, 2010). Changes in the physical properties of the atmosphere caused by an increase in greenhouse gases may result in different temperature responses in low-lying compared to mountainous areas (Pepin et al., 2022). Consequently, meteorological station networks with dense temporal and spatial coverage are required to capture the complex temperature patterns attributed to local factors like surface conditions (albedo, roughness, topographic aspect), cold air pooling, and temperature inversions (Rolland, 2003). In addition, knowledge about the atmospheric conditions such as synoptic-scale circulation patterns, wind regimes, and atmospheric moisture is required to explain the observed variations in lapse rates.

Trends and variability in surface air temperature are known to be larger in the Arctic region than on the global average, a phenomenon commonly referred to as 'Arctic amplification' (Serreze et al., 2009). This is largely driven by the loss of the sea ice, followed by oceanic heat gain through the ice-albedo feedback, and infrared radiation feedbacks (Serreze et al., 2009; Screen and Simmonds, 2010; Bintanja and Van Der Linden, 2013; Ono et al., 2022). Observational studies of Greenland's temperature time series show a significant coastal warming (Hanna et al., 2021), with pronounced warming on the west coast during the winter (Abermann et al., 2017; Hanna et al., 2012) and weaker but steady warming during the summer months both in West and East Greenland (Abermann et al., 2017; Mernild et al., 2014). During the period 2013–2017, the rate of temperature increase accelerated especially in the Northeast and North Greenland (Jiang et al., 2020). Moreover, model results suggest that Northeast Greenland is among the most sensitive areas to changing temperature (Schuster et al., 2021; Bintanja and Selten, 2014; Shepherd, 2016). With respect to elevation dependency of temperature trends in the Arctic, Gardner et al. (2009) suggest that under a warming climate near-surface temperature lapse rate decreases.

As near-surface air temperature is influenced by various surface processes (Minder et al., 2010; Pepin and Seidel, 2005), temperature lapse rates are not linear and differ from the vertical environmental lapse rate (ELR) of 6.0–6.5 °C km$^{-1}$. Thus, one must differentiate between the vertical ELR and the near-surface temperature gradient (Heynen et al., 2016; Thayyen and Dimri, 2018). While the former is mainly controlled by adiabatic effects and stratification, the latter is more strongly influenced by the local surface energy balance (net radiation and turbulent heat fluxes), (Cullen and Marshall, 2011; Marshall et al., 2007; Gao et al., 2012). In this study, we mainly focus on the latter (near-surface temperature gradient) and investigate its variation as well as its drivers.

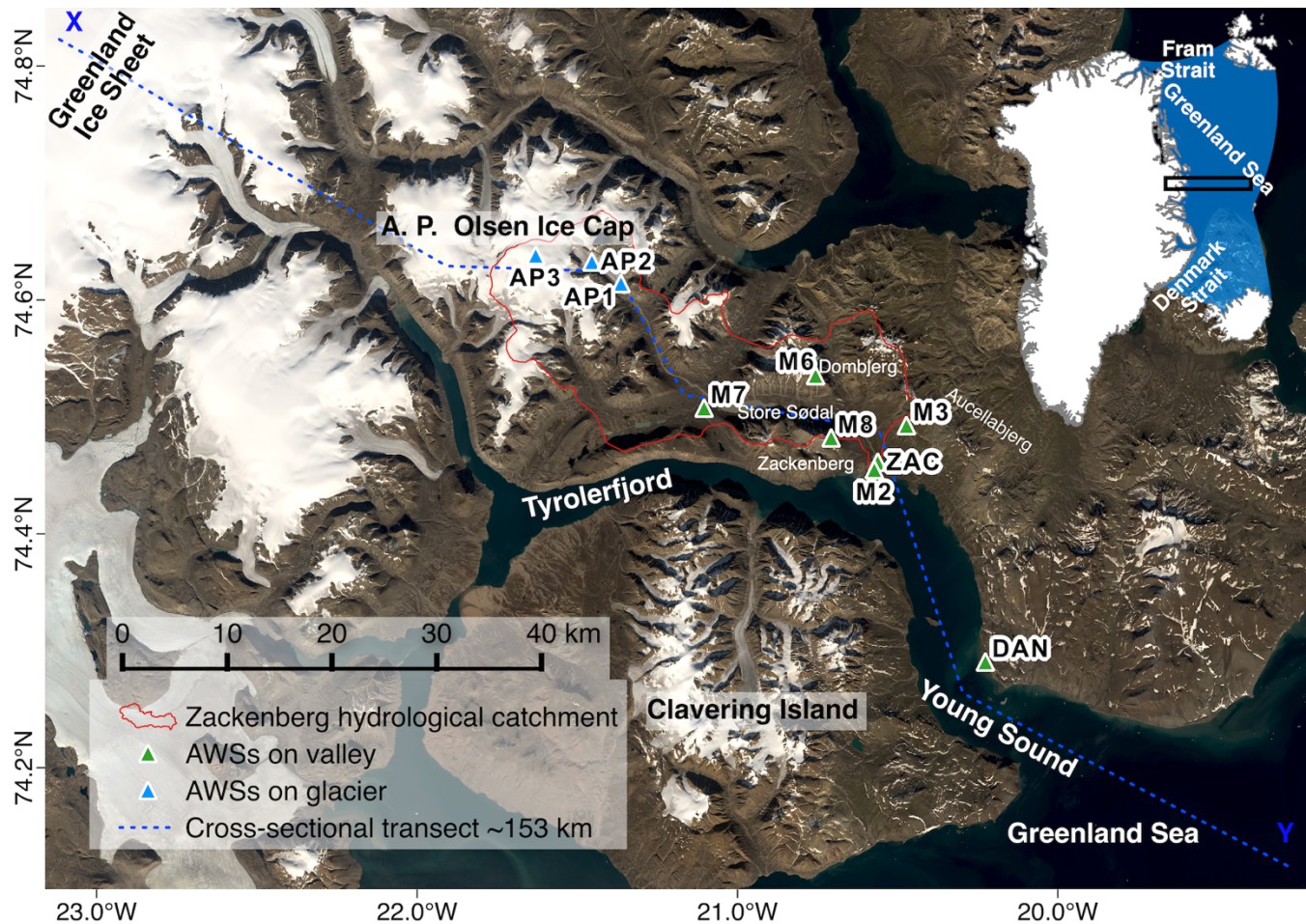

**Figure 1. Map of the Zackenberg region (ZR).** The location of automated weather stations (AWSs) used in this study in the Zackenberg catchment (red contour) distributed at different elevations and surface types are shown with a triangle; the green triangle indicates AWSs on the ice-free surface and the blue triangle on the glacier surface. XY (blue) dashed line represents the cross-sectional transect of ~153 km in length passing from the Greenland Ice Sheet (GrIS) (from the west) towards the Greenland Sea (to east) along the ZR and Young Sound. The vertical temperature profiles were taken by unmanned aerial vehicles close to the ZAC station. The inset image shows the location of the ZR in Greenland (within a black rectangle) and the Greenland Sea (~$1.2\times10^{6}$ km$^2$, blue shade) located at the east of Greenland and west of the Svalbard archipelago. The black rectangle indicates an area that is zoomed-in for the purpose of studying sea-ice-related terrestrial local-scale anomalies in the follow-up figures. The (zonal) length of the black rectangle represents the distance from the GrIS (to the west) to the average sea-ice extent for the summer season (to the east), while its (meridional) width covers the ZR. The Greenland Sea is defined by the International Hydrographic Organization (available online at http://www.marineregions.org/). The base image is Landsat 8 imagery retrieved on 2017-08-24

To study and understand climate variability in the coastal area of Greenland, either dense meteorological station networks or high-resolution atmospheric model simulations are needed. We use both datasets; thus, we can utilize climate information in the large domain. For Greenland, such a dense station network is only available for the Zackenberg region (ZR), (Northeast Greenland), serving as a "research laboratory", as part of the Greenland Ecosystem Monitoring (GEM) program (https://g-e-m.dk), (Fig. 1; see Sect. 2.2). The unusually high spatial resolution of observations in the ZR provides

an opportunity to combine data sets to gain a better understanding of how an Arctic environment responds to a changing climate and the related feedback mechanisms. Abermann et al. (2017) found weaker but statistically significant climate trends in Zackenberg compared to overall Greenland climate trends. They suggested that changes in this region's ecosystem due to climate drivers might thus be weaker than changes in more sensitive areas of Greenland.

Previous studies showed a positive correlation between the temperature in Greenland and persistent, strong 500 hPa anticyclones over Greenland, which are commonly quantified through the Greenland Blocking Index (GBI), (Hanna et al., 2021; Jiang et al., 2020; Ballinger et al., 2018a, b). Large positive GBI values are often connected with atmospheric blocking events (Hanna et al., 2016). During the blocking event, the polar jet streams develop nearly stationary meanders and trap warm air in an anomalously poleward area (Sirpa et al., 2011; Hanna et al., 2015). Atmospheric blocking, depending on the
location of the block, can advect warm air across Greenland (especially over western Greenland), (Fettweis et al., 2011, 2013; Hanna et al., 2016, 2014) or stimulate adiabatic descent (or large-scale subsidence) and warm the low-tropospheric air (Ding et al., 2017; Hofer et al., 2017). Since the GBI merely measures the average geopotential height at 500 hPa pressure level (Z500) over Greenland, not the types or locations of the blocking anticyclones (Preece et al., 2022), it can affect regional climates differently. For instance, Preece et al. (2022) found cyclonic wave breaking patterns (a type of blocking that occurs when the anticyclone approaches Greenland from the east along the poleward flank of a cutoff low) produce
more melt in Northeast Greenland. This underscores the need to analyze the potential connection between atmospheric circulation and climate variability in topographically complex regions like ZR, which to our knowledge has not been undertaken in the past.

Sea ice is known to be a relevant driver of the atmospheric state in nearby regions in the Arctic (Stroeve et al., 2017, 100     2014; Müller et al., 2021; Schuster et al., 2021; Stranne et al., 2021; Isaksen et al., 2022). This is relevant for Zackenberg as it is situated inside a protected fjord system close to the coast (Fig. 1). Yet it exhibits a continental climate with very cold winters and low precipitation due to a wide belt of sea ice. Local climate change in Zackenberg will therefore be determined by the sea-ice variability (Stendel et al., 2008). Variations in sea-ice export through the Greenland Sea, as well as the North Atlantic Oscillation (NAO), have an important impact on the spatiotemporal variability of the climate in the ZR, especially
the precipitation and surface temperature. Reduction in sea ice leads to an increase in precipitation and temperature (Hinkler, 2005; Hansen et al., 2008). However, the direct observational linkages between the sea-ice variability and terrestrial local temperature gradients along the mountain slope in the ZR have not been analyzed.

In summary, a recent increase in research into fundamental processes in Northeast Greenland has arisen from the region's sensitivity to global climate change. Despite this, there exists a distinct knowledge gap, especially regarding the
consistency of the observed spatial and temporal variability of temperature gradients in the orographically complex terrain, for which the ZR is a useful showcase. To close this gap, this study aims to better understand the coastal near climate variability in Zackenberg, mainly looking at 2 m air temperature ($T_{2m}$), and the relevant driving mechanisms using a dense network of automated weather stations (AWSs) and combining data from atmospheric reanalysis and a regional climate model. To achieve this general aim, the paper has the following objectives: (1) to estimate the role of surface conditions,

general atmospheric circulation, and weather conditions in controlling vertical differences in the change of air temperatures, and hence the slope temperature gradient (STG; see Sect. 2.8); (2) to diagnose the physical causes of STG variability due to sea-ice change; (3) to discuss the implication of STG and sea-ice variability on the surface mass balance (SMB) of a local ice cap, A. P. Olsen Ice Cap (APO), (Fig. 1). Finally, the impact of the temporal evolution of sea ice on the atmospheric processes are accounted. By doing so, the present study examines the statistical relationship between STG, sea ice, and other

atmospheric variables and their physical connections.

## 2 Data and methods

### 2.1 Physical setting/site description

The long-term (1996–present) spatiotemporal air temperature monitoring program at the ZR (74.5° N, 20.5° W) made this region a good choice for our study. To investigate the relevant drivers of the STG variability in the ZR, areas toward the

Greenland Sea as well as towards the Greenland Ice Sheet (GrIS) are included as regions of interest (Fig. 1). The ZR is located in Northeast Greenland, midway between the outer coast (~40 km to the east of ZR) and the GrIS (~70 km to the west of ZR). The area shows a high degree of complexity with respect to mountain orography (reaching up to 1450 m a.s.l.) and a wide valley reaching sea level and joining fjords (Young Sound and Tyrolerfjord). A network of AWSs measuring meteorological variables is installed within the 2–3 km wide valley. The location of the AWSs used is shown in Fig. 1. The

longest-running AWS is ZAC, which was established by ClimateBasis monitoring programme (GEM, 2020a) in August 1995 (Fig. 1). Based on the data from ZAC, the monthly average $T_{2m}$ is highest in July (+6 °C) and lowest in February (-20 °C), based on the period 1996–2019. The mean total annual precipitation is 204 mm for 1996–2020. The area has a polar night for ~89 days (7 November–3 February) and a polar day for ~106 days (30 April–13 August), (Hansen et al., 2008). The Zackenberg River catchment covers 514 km$^2$, 20 % of which is glacierized (Meltofte and Rasch, 2008).

APO (74.6° N, 21.5° W) is the largest ice cap in the Zackenberg River catchment, of which the southeast-flowing outlet glacier is a key contributor to the meltwater (Mernild et al., 2008), (Fig. 1). The glacier is situated around 35 km northwest of ZAC, disconnected from the GrIS. It is also instrumented with AWSs as a part of GEM (Fig. 1).

Air temperature inversions, atmospheric configurations where temperature increases with height, occur regularly in the ZR (Hansen et al., 2008). For the period 2004–2018, in between ZAC (43 m a.s.l.) and M3 (420 m a.s.l.), (where M3 is

representative of the area above the inversion layer), inversions are frequent (70–75 %) during winter, whereas during summer the inversion frequency decreases (27–58 %). This local-scale seasonal inversion pattern is consistent with a recent reanalysis-based study across Greenland (Shahi et al., 2020).

### 2.2 Automated weather stations data

**Table 1. Overview of all automated weather stations used in this study covering different surface types**

| Monitoring programme/ institution | Station | Surface type | Latitude (° N) | Longitude (° W) | Elevation (m a.s.l.) | Operation period |
|---|---|---|---|---|---|---|
| ClimateBasis | ZAC | | 74.47 | 20.55 | 43 | 1995– |
| GeoBasis | M2 | Tundra | 74.46 | 20.56 | 17 | 2003– |
| | M3 | | 74.50 | 20.46 | 420 | 2003– |
| | M6 | Rocks and boulders | 74.54 | 20.74 | 1282 | 2006–2010 |
| | M7 | | 74.51 | 21.10 | 145 | 2008– |
| | M8 | | 74.49 | 20.70 | 1144 | 2013– |
| GlacioBasis | AP1 | Ice/Snow | 74.62 | 21.37 | 660 | 2008– |
| | AP2 | | 74.64 | 21.47 | 880 | 2008– |
| | AP3 | | 74.64 | 21.65 | 1475 | 2009– |
| DMI | DAN | Tundra | 74.30 | 20.21 | 44 | 2009–2016 |

This study uses the meteorological data measured by several AWSs operating under GEM. It is a long-term research program on climate change effects and ecosystem interaction initiated in 1995 and has been collecting long-term interdisciplinary environmental datasets at Zackenberg (Meltofte and Thing, 1996; Elberling et al., 2008; Olesen et al., 2010). We use meteorological data from three sub-programs: ClimateBasis (ZAC), (GEM, 2020a), GeoBasis (M2, M3, M6, M7, and M8), (GEM, 2020b, c, d, e, f), and GlacioBasis (AP1, AP2, and AP3), (GEM, 2020g, i, h), (Fig. 1 and Table 1).

These AWSs are distributed at different elevations and over different surface types: while three AWSs are over ice, six AWSs are over gravel or tundra surfaces in the ZR (Table 1). Additionally, we use meteorological data from station Daneborg (DAN), located 23 km east of ZAC near the outer coast (Cappelen et al., 2001), operated by the Danish Meteorological Institute (DMI), (Fig. 1 and Table 1). These datasets provide excellent spatial and vertical coverage for the investigation of inversion layer formation and destruction processes that occur near the surface. The dataset from the

monitoring programs is comprehensively described in Jensen et al. (2014) and as metadata accompanied by the GEM database. All stations are equipped with sensors for $T_{2m}$, relative humidity, wind speed (U) at 2m ($U_{2m}$) and direction, and the surface pressure ($P_{surf}$). A smaller set of stations is also equipped with radiometers and snow depth sensors (Table S1). These stations have a variety of exposure relative to the surrounding terrain and many of the low-elevation valley stations are likely to at least occasionally reside in localized cold air pools.

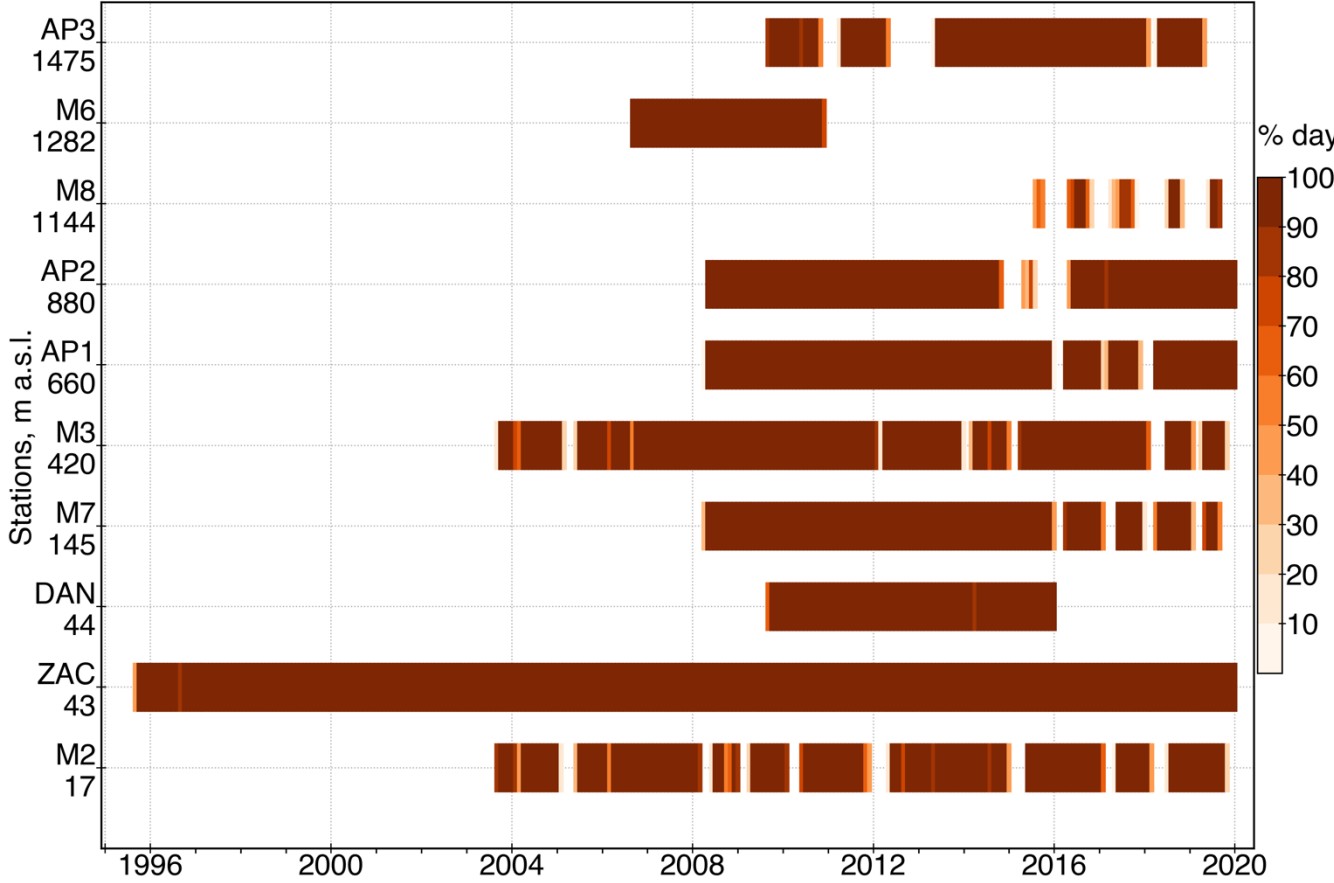

**Figure 2. Temporal coverage of all AWSs located in the ZR and on the APO and the elevation of each station in meters above mean sea level (m a.s.l.). The color bar indicates the percentage of days with air temperature data during a respective month for each AWS after screening**

The following quality control procedures were applied to the meteorological data: first, we manually corrected the data by removing physically unrealistic outliers by visualization of the time series. Second, we calculated the daily means provided that at least 80 % of the hourly measurements were recorded (Abermann et al., 2017). Similarly, monthly means are calculated if more than 80 % of the daily means exist. Figure 2 shows the percentage of days with $T_{2m}$ data during a respective month for each AWS after screening.

The relative humidity was corrected for unrealistic measurements below 0 °C to values for ice using the method proposed by Anderson (1994). As a result, the relative humidity with respect to liquid water below 0 ºC measurements were multiplied by the ratio of the saturation pressure of water and ice.

## 2.3    Derived variables

The 2-m specific humidity ($q_{2m}$) was calculated from relative humidity and saturation specific humidity. Firstly, we calculated the saturation vapor pressure over ice and water following Goff & Gratch (1946). Secondly, using the saturation

vapor pressure values, saturation specific humidity was calculated (Fausto et al., 2021). Finally, the specific humidity was estimated using relative humidity and saturation specific humidity.

For all the sites, we estimated the longwave-equivalent cloud cover fraction based on the strong relationship between $T_{2m}$ and the incoming longwave radiation ($LWR_{in}$), (Van As, 2011). For this, we used the approach developed by Swinbank (1963) to calculate a theoretical $LWR_{in}$ corresponding to both clear-sky conditions as a function of $T_{2m}$, and overcast conditions assuming blackbody radiation. The cloud cover fraction (CCF) is then calculated by linear interpolation of the $LWR_{in}$ between the clear-sky and overcast estimates. Essentially, the CCF computed with observed variables is closely associated with sky emissivity, rather than the physical fraction of the sky covered by clouds (Djoumna et al., 2021). See Van As (2011) for more details on the CCF estimation.

## 2.4    Fjord-ice conditions

We use information about the fjord-ice formation and break-up period in the outer Young Sound (Fig. 1) from the MarineBasis sub-program of GEM (GEM, 2020j). The dates for the onset and breakup of fjord-ice in outer Young Sound are mainly based on an automatic digital camera system (74.3° N, 20.2° W), (Rysgaard et al., 2009). The camera is programmed to take one photo a day at 13:20 all year round. In the years when the camera malfunctioned (only a few times) the data are validated with satellite observations.

## 2.5    Event-based vertical profiling with unmanned aerial vehicle platform and iMET-XQ2 sensor

An optimal combination of unmanned aerial vehicles (UAVs) and small meteorological sensors provides the potential for measurements of atmospheric variables in a high spatio-temporal resolution (Kimball et al., 2020). Since UAV provides a dynamic and flexible platform for meteorological data acquisition, it has been used widely in exploring atmospheric boundary layer conditions and features, such as inversions and vertical profiles (Kimball et al., 2020; de Boer et al., 2020; Hemingway et al., 2017).

In the summers of 2017 and 2018, field campaigns were carried out in order to measure high-resolution vertical air temperature profiles within 10 m from the ZAC using the International Met System iMET-XQ2 (iMet-XQ2 Second-Generation Atmospheric Sensor for UAV Deployment, 2021) sensor mounted onto a DJI MAVIC Pro multirotor aircraft (location of the flight is indicated in Figure 1). Table S2 shows the specifications of the temperature sensor and GPS used for the measurements. The iMET-XQ2 is a self-contained sensor package designed for UAVs to measure atmospheric temperature, pressure, and relative humidity. It is also equipped with a built-in GPS and an internal data logger along with a rechargeable battery.

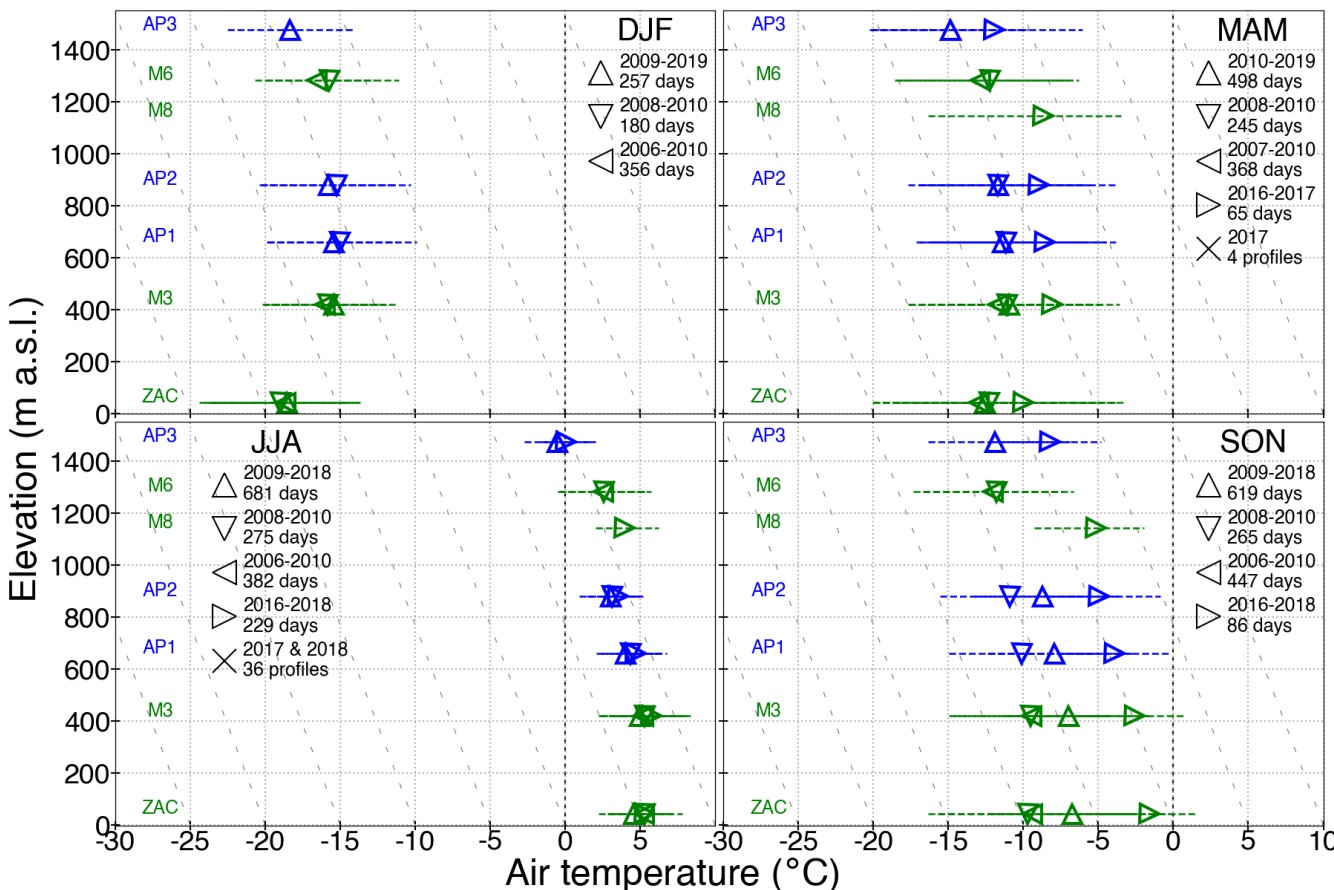

**Figure 3. Mean air temperature measurements from AWSs and UAV against the elevation. Each marker (except for the cross) represents the daily mean temperature of a station for temporally overlapping periods, as shown in the legend, with other AWSs measurements for each season. The 25th and 75th percentiles of air temperature are indicated by the dashed horizontal line passing across the marker. Blue and green colors represent stations on ice and ice-free surface, respectively. The mean vertical temperature profile measurements from UAV taken in 2017 (black line) and 2018 (solid grey line) are shown (in MAM and JJA panel) after applying Savitzky Golay low-pass filter to remove the high-frequency fluctuation of air temperature due to the high temporal resolution of UAV. The shaded area around the line represents the 25th and 75th percentiles of the temperature at the given elevation from all vertical profiles. The 'X' marker represents the sub-daily temperature of a station temporally overlapping with the UAV flight period. In each panel the mean environmental lapse rate (dashed grey line) of 6.5 ºC km⁻¹ is plotted for reference**

In total, we used 40 temperature profiles (16 from 2017; 24 from 2018), (Figs. 3 and S1). Since the vertical resolution of the temperature profiles measured by UAV is high, we first averaged the data over the height intervals of 1 m, and then, smoothed it at 25-m interval using the Savtizky Golay low-pass filter (Savitzky and Golay, 1964), considering the vertical accuracy of GPS (12 m; Table S2). Air temperature measurements from the UAV at the same time and height as ZAC and M3 were extracted for comparison (Figs. 3 and S1).

## 2.6    ERA5

We use the fifth-generation atmospheric reanalysis ERA5, a product of the European Centre for Medium-Range Weather Forecasts (ECWWF) from 1979 onwards (Hersbach et al., 2020). In particular, we use the daily mean upper-air variable such as Z500 (Hersbach et al., 2018b, a). Additionally, the daily fractional sea-ice cover (SIF) from ERA5, interpolated and prescribed to the Regional Atmospheric Climate Model (RACMO) grid at 5.5 km (detailed in Sect. 2.7), was used.

ERA5 has a horizontal resolution of approximately ~0.25° (~15 km over the study region), with 137 model levels, from the surface up to 0.01 hPa (~80 km). It shows many improvements compared to its predecessors (ERA-15, ERA-40, and most recently ERA-Interim) in converting the model values to observation equivalents, and in the processing of observations in the Integrated Forecasting System (IFS Cy41r2 4D-Var), (Graham et al., 2019; Delhasse et al., 2020). In addition, various newly reprocessed datasets have been assimilated and show consistent sea surface temperature and sea ice (Hersbach et al., 2019).

The SIF data are derived from satellite passive microwave brightness temperatures from the series of SMMR, SSM/I, and SSMIS sensors and are prescribed in ERA5 (along with sea surface temperatures) to provide sea surface boundary conditions for the ERA5 atmospheric model. The following processing steps are considered when these products are ingested into ERA5: (1) removal of <15 % SIF; (2) interpolation to the ERA5 model grid; (3) prescribing ice-free conditions for sea surface temperature >3 °C (European State of the Climate 2019).

## 2.7    The Regional Atmospheric Climate Model

We use the polar version of the Regional Atmospheric Climate Model (RACMO), version 2.3p2, with a horizontal resolution of 5.5 km for the period 1990–2020 (Van Meijgaard et al., 2008; Noël et al., 2015). The polar version of RACMO was developed by the Institute for Marine and Atmospheric Research (IMAU), Utrecht University, with the primary aim to represent the evolution of SMB of glaciated regions such as Greenland (Noël et al., 2015). RACMO encompasses the dynamical core of the High Resolution Limited Area Model (HIRLAM) and the atmospheric physics module from the ECMWF IFS (Noël et al., 2015). At the lateral atmospheric boundaries, the model is forced by ERA5 (1990–2020) at a 6-hourly time interval within 24-grid cells wide relaxation zone (Noël et al., 2019). At each of the 40 vertical atmospheric model levels, the following atmospheric forcing is prescribed: temperature, specific humidity, pressure, wind speed, and direction. Furthermore, the daily SIF and sea surface temperature are not generated by RACMO itself but are prescribed from ERA5 (1990–2020), after being interpolated onto the RACMO model grid (5.5 km). For a detailed model description, we refer to Noël et al. (2019).

We use the daily variables of air temperature and specific humidity at 1000 hPa, 925 hPa, 850 hPa, 700 hPa, and 500 hPa pressure levels from RACMO. The daily near-surface variables used include $T_{2m}$, $q_{2m}$, $P_{surf}$, U at 10 m ($U_{10m}$), and vertically integrated cloud content, i.e., ice (IWP) and liquid (LWP) water paths.  The daily SIF interpolated to the RACMO grid is also used in this study.

Noël et al. (2019) evaluated the simulated GrIS climate using in situ measurements; they found that RACMO closely represents the near-surface climate ($0.73 < R^2 < 0.98$) at 37 AWSs. However, the near-surface climate evaluation done by Noël et al. (2019) does not cover stations on the ice-free surface, since RACMO was developed for studies focusing on the GrIS. Since most of the AWSs in our study domain are on the orographically complex ice-free surface (Fig. 1), a thorough evaluation of RACMO is a crucial prerequisite.

For model evaluation, we used thoroughly quality-controlled daily meteorological data as described in Sects. 2.2 and 2.3, and the temporally overlapping RACMO subsampled grid cell nearest to the observation site (Fig. S2 and Table S3). Unlike Noël et al. (2019), we did not use the elevation criteria (<100 m) to select the nearest grid cell. Table S3 shows that RACMO bias varies amongst variables and stations. The modeled and measured $T_{2m}$ and $q_{2m}$ agrees well for all stations in all seasons ($0.5 < R^2 < 0.9$). However, biases are smaller for stations on the ice compared to gravel or tundra surfaces. A bias for $P_{surf}$ is found, which evidently is the result of the elevation difference between the station and the model (Noël et al., 2019). Discrepancies in modeled $U_{10m}$ and observed $U_{2m}$ can be partly attributed to sensor uncertainty, modeled $P_{surf}$, and the height at which U is measured (2 m) and modeled (10 m). Despite the elevation and pressure bias, RACMO can resolve the local wind direction (i.e., sea breeze; Fig. S3) in the ZR, especially during summer. Overall, discrepancies exist between model and observations, however, RACMO represents the seasonal pattern appropriately, which is the basis for composite analysis (see Sect. 2.9).

We also used a 1-km SMB product, statistically downscaled from the output of the regional atmospheric climate model RACMO 2.3 (resolution of 11 km), (Noël et al., 2017). Noël et al. (2017) evaluated the 1-km SMB product against observed data for the APO and found close agreement between the downscaled SMB and observations ($R^2=\sim0.7$). Additionally, 1-km air temperature and snowfall data for the APO are also used.

## 2.8 Slope temperature gradient

Following Glickman (2000), the lapse rate is the change of an atmospheric variable with height, the variable being temperature, unless otherwise specified. On average, temperature decreases with elevation above sea level to the tropopause at the rate of ~6.5 ºC km$^{-1}$—commonly referred to as the vertical environmental lapse rate. However, the temperature gradient along the mountain slopes significantly varies from the environmental lapse rate (Heynen et al., 2016; Thayyen and Dimri, 2018). This is due to the fact that orographic effects and surface processes strongly control the temperature regime, and hence, alter the near-surface boundary layer (Ayala et al., 2016). In this study, the temperature gradient along the mountain slope is termed slope temperature gradient (STG). This temporally and spatially varying STG (°C km$^{-1}$) is calculated as,

$$STG_{US-LS} = \frac{T_{US} - T_{LS}}{z_{US} - z_{LS}}$$

where T and z are mean temperatures and elevations, respectively, of lower station (LS) and upper station (US). The station pairs used to calculate STG are indicated as a subscript unless otherwise stated; for instance, the $STG_{M3\text{-}ZAC}$ denotes the STG between M3 (US) and ZAC (LS).

Reference to an increase or decrease in the lapse rate terminology can lead to ambiguity. To avoid confusion, we follow the terminology proposed by Pepin and Losleben (2002), where a 'steep' STG is a rapid decrease of temperature with elevation, and a 'shallow' STG is a less negative or positive STG (i.e., temperature inversions). The terminology was used as follows: for a given STG, steeper STG < STG < shallower STG. For example, if the ELR is -6.5 °C km$^{-1}$ and a STG between AP2 and AP1 stations ($STG_{AP2\text{-}AP1}$) is -7 °C km$^{-1}$, and between M3 and ZAC stations ($STG_{M3\text{-}ZAC}$) is -5 °C km$^{-1}$ then ELR is shallower than $STG_{AP2\text{-}AP1}$ whereas steeper than $STG_{M3\text{-}ZAC}$. Additionally, we refer to more and less positive STG as 'strong' and 'weak' inversions, respectively.

## 2.9    Composite analysis technique

### 2.9.1    Based on the slope temperature gradient

We employed a composite analysis technique to scrutinize the link between STG, and near-surface and synoptic-scale climate variables to avoid the assumptions corresponding to linear correlation analysis between these variables (Schweiger et al., 2008; Laken and Čalogović, 2013). This technique highlights the low-amplitude signal within data which otherwise would be muted due to background variability (Laken and Čalogović, 2013). In general, this method is more adequate than the assumption of a linear relationship. The subset of data is selected based on objective criteria e.g., days with shallow STG, and then averaged (Forbush et al., 1983). Consequently, low-amplitude signals are isolated and become visible due to the reduction in the stochastic background variability (Laken and Čalogović, 2013). The resulting composite anomaly can emphasize the potential relationship between atmospheric phenomena and the variables used for the composite segregation.

The composites of atmospheric and surface variables were computed corresponding to days with high and low STG in Zackenberg. Let $STG_{US-LS}(d_{all})$ be the daily STG between a pair of stations for the given season with $d_{all}$ number of days; $STG_\mu$ and $STG_\sigma$ are the temporal mean and standard deviation, respectively, of $STG_{US-LS}(d_{all})$. Based on that, we classified days as high and low STG days. For high STG days, let $d_+$ be the selected days in the given season that fulfill $STG_{US-LS}(d_+) > STG_\mu + 0.5\ STG_\sigma$. Likewise, for low STG days, let $d_-$ be the selected days in the given season that fulfill $STG_{US-LS}(d_-) < STG_\mu - 0.5\ STG_\sigma$. Finally, we computed high ($V_+$) and low ($V_-$) composite means for the variable V (e.g., SIF) at geographic location (lon, lat) as,

$$V_+(lon, lat) = \frac{1}{N_+} \sum_{i_+} V(lon, lat, d_+)$$

$$V_-(lon, lat) = \frac{1}{N_-} \sum_{i_-} V(lon, lat, d_-)$$

where $N_+$ and $N_-$ are the total number of high and low STG days, respectively. Since the selection of composites is based on a unidimensional variation of STG (i.e., it only varies temporally), all grid cells of the composited variables (e.g., SIF, which varies in space and time) contain means over the same subset of days (i.e., $d_+$ and $d_-$ are same for all grid cells).

To ensure the robustness of the applied method and reproducibility of the results, we used several other criteria and cross-checked the computed composites. First, we used STG calculated from various pairs of stations to compute corresponding composited variables following the aforementioned criterion. Additionally, we used criteria based on the 25th and 75th percentiles of the STG distribution in order to isolate high and low STG days, respectively. This was done to objectively evaluate the potential of the applied threshold (i.e., $STG_\mu \pm 0.5\ STG_\sigma$) to get the low amplitude signal on two extreme sides of the data spectrum (~66 % of data). Furthermore, the representativeness of the sample size was tested by using several sample sizes of STG based on all available days, exclusively inversions (positive shallow STG) and non-inversions (steep STG) days, and compare the anomaly patterns of the composited variable. Also, STGs, calculated from different pairs of temporally overlapping stations, were used to compute the composites, to check the consistency of the results produced by different station pairs for the same sample size. Since all the aforementioned criteria show similar anomaly patterns (see Sect. 3.2), all results based on $STG_\mu \pm 0.5\ STG_\sigma$ and for all available days for STG variations will be shown.

### 2.9.2 Based on sea ice

To investigate the possible link between sea ice, near-surface and upper-air atmospheric conditions, and consequently STG, we applied the following criteria in defining high and low SIF days. The criterion is similar to the one used for STG, however since SIF is typically not normally distributed, 95th and 5th percentiles were used to discern high and low SIF days.

Let $SIF(d_{all})$ be the daily SIF for the given season with $d_{all}$ number of days; $SIF_{95th}$ and $SIF_{5th}$ are the 95th and 5th percentiles, respectively, of $SIF(d_{all})$. High and low SIF days are the days when $SIF(d_+) > SIF_{95th}$, and $SIF(d_-) < SIF_{5th}$, respectively. When $SIF_{5th}$ is equal to zero, all days with zero SIF are considered low SIF days.

Two sets of composites were calculated based on SIF: the first set of composited variables is based on local information of SIF from the same grid cell, and thus, includes regions only covered by sea ice. This approach increases the probability of identifying signals in atmospheric phenomena due to local changes in sea ice. The second set of composite variables is based on a daily time series (1996–2020) of zonally (overall grid cells) averaged (median) SIF over the Greenland Sea (Figs. 1 and S4). Thus, the corresponding composited variables include inland regions beyond sea-ice coverage. The first set of composited variables shows the spatial heterogeneity of composite anomalies due to SIF variability within the same grid cell. Since these composites are based on SIF from the same grid cell, bordering grid cells do not essentially contain the average over the same subset of days (i.e., $d_+$ and $d_-$ might not be the same for all grid cells). In contrast, since the second set of composites is based on a unidimensional variation of zonally averaged SIF over the Greenland Sea, all grid cells of

composited variable essentially contain the average over the same subset of days (i.e., $d_+$ and $d_-$ are the same for all grid cells).

To account for the impact of the climatological evolution of sea ice on the interpretation of the composites we used a nonparametric kernel change point detection algorithm to detect an abrupt change (change point) in the statistical properties
of the annual average sea ice time series (1996–2020), (Celisse et al., 2018; Arlot et al., 2019; Truong et al., 2020), (Fig. S5). We used the radial basis function kernel. This method transforms the data to a higher-dimensional space, called a reproducing kernel Hilbert space in which simple linear models are trained to produce analytical, low bias, and low variance models. Given the model, it computes the optimal change points that minimize a certain cost function i.e., the goodness-of-fit of the subset of time series to a specific model. See Truong et al., (2020) or Celisse et al., (2018) for a detailed exact
optimization procedure. The seasonal average rather than daily sea ice data (1996–2020) was used for detecting the change points. This was done to avoid detecting false positive change points because of noise in the data.

Figure S5 shows the detected change points for each season. We found two change points per season: for winter 2003 and 2014, for spring 2002 and 2015, for summer 2002 and 2008, and for autumn 2001 and 2008. Interestingly, the early 2000s contain a consistent change point in all the seasons, the mid-2010s in winter and spring, and 2008 in summer and
autumn. Based on the detected change points for all the seasons and on account of consistency, we chose 2002, 2008, and 2014 as change points for all seasons. It has shown that the sea ice loss accelerated from 2000 onwards, (Ivanov, 2023; Stroeve and Notz, 2018). Furthermore, according to NSIDC (National Snow and Ice Data Center, 2023), the 2008 minimum sea-ice extent is the second-lowest recorded since 1979 (2007 being the lowest). The 2012 minimum sea-ice extent was recorded lowest of the satellite-era (Perovich et al., 2013); this might have a seasonally lagged impact. Hence, these studies
provide confidence in the detected change points.

To ensure the stationarity of the subsets of the sea ice time series, we used a combination of one parametric test i.e., the Augmented Dickey-Fuller test (ADF), (Dickey and Fuller, 1979; Said and Dickey, 1984), and two nonparametric-tests i.e., Kwiatkowski–Phillips–Schmidt–Shi (KPSS), (Kwiatkowski et al., 1992), and Phillips-Perron test (PP), (Phillips and Perron, 1988). Inference as to whether a given time series is stationary or not was made based on all three stationarity tests to help
against the confirmation bias. The results are shown in Figure S5. All the subsets of the time series are inferred stationary except for the early 2000s period (1996–2001) for all seasons and the period 2001–2007 for autumn. However, the result of the KPSS test which is more powerful than the other two tests in the small samples (Arltová and Fedorová, 2016) indicates that we cannot reject the null hypothesis and infer that the subset of the time series is stationary.

Hence, in addition to the period 1996–2020, we divided the daily sea ice time series into the following subperiods to
perform the composite analysis: 1996–2001, 2002–2013, 2014–2020, 2002–2007, and 2008–2020. The computed composites were cross-checked to account for the impact of the temporal evolution of sea ice on the interpretation of the atmospheric processes.

We applied the nonparametric Wilcoxon rank-sum test to detect statistically significant differences between high and low composites of STG and SIF. This method tests at each grid point the local null hypothesis $H_o: V_+(lon, lat) =$

$V_-(lon, lat)$ at a 95 % confidence level.

## 3 Results

The changes in $T_{2m}$ and the resulting STG can be attributed to many different drivers. Here, we explored the seasonality of the STG in the ZR and the variables influencing the STG. Along with the local factors like surface type and fjord-ice condition, STG anomaly is triggered by various large-scale factors.

### 3.1 Slope temperature gradient

The near-surface climate is influenced by its surface characteristics (e.g., the occurrence of snow, bare soil, vegetation, etc). The inclusion of the UAV-based air temperature profiles, which represent the vertical temperature gradient, helps to discern a comprehensive perspective of temperature variability in the ZR and complements the results. Figure 3 gives a general overview of seasonal mean temperature profiles from AWSs and UAV for temporally overlapping periods between the

stations. Note, that all UAV-based measurements are made during the summer months. During the UAV measurement period in the summer of 2017, the valley was covered by snow, and foggy conditions prevailed, whereas, during the measurement period in the summer of 2018, there was scattered snow cover. The average temperature profile from the UAV measurements in the summer of 2017 is marked by the presence of strong inversions, and it is distinct from the rest of the average summer temperature profiles from the AWSs (Figs. 3 and S1).

The ZAC and M3 stations were used for comparison of the sub-daily vertical temperature gradient from the UAV and STG (Fig. S1). There is a close correspondence between the UAV and the AWS temperature measurements at the same elevation (ZAC, 43 m a.g.l.; M3, 420 m a.g.l.), ($R^2$ = 0.8–0.9, p < 0.05; Fig. S1). Notably, the temperature difference (M3 minus ZAC) calculated from AWSs and the corresponding UAV-based measurements show a significant correlation ($R^2$ = 0.7–0.9, p < 0.05) with the mean absolute error (MAE) of 0.6–1.0 °C. This highlights the potential of AWSs to capture the

inversions in the ZR, despite the surface characteristics influencing screen-level (2 m) temperature.

Nonlinear temperature profiles are evident during the winter season, in which temperature increases vertically at lower elevations (i.e., ~43–420 m a.s.l., ZAC–M3) indicating the presence of inversions and then gradually decreases further up (Fig. 3). During winter, at the APO (where AP1, AP2, and AP3 are located), the temperature decreases with elevation and the STGs are consistently shallower than the vertical ELR ($STG_{AP2-AP1}$ = -2 °C km$^{-1}$ and $STG_{AP3-AP2}$ = -5 °C km$^{-1}$ versus -6.5

°C km$^{-1}$).

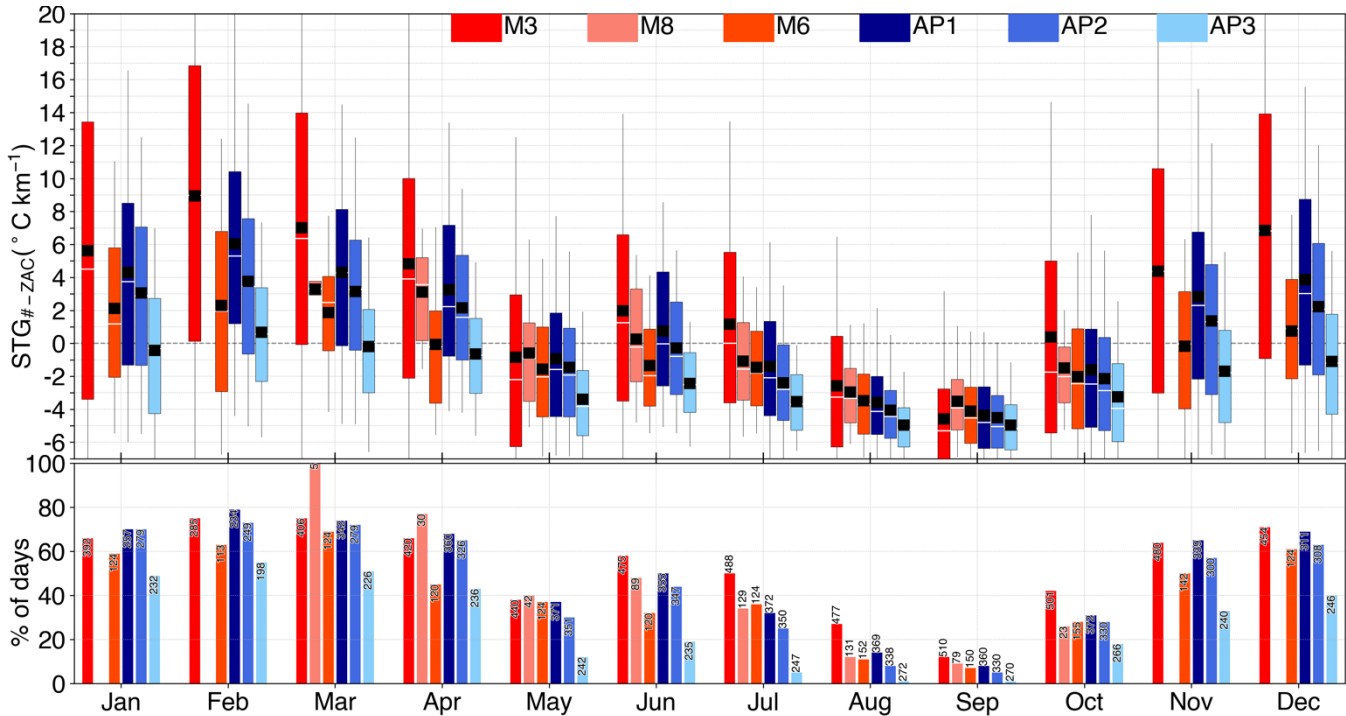

**Figure 4. Annual cycle of monthly STG (°C km⁻¹) between ZAC and other higher AWSs (# indicates stations listed in the legend). The upper panel shows the boxplot of daily mean STG for each month, and the lower panel shows the frequency (% of days) of inversions in total measurement days (number inscribed on or at the top of the bar plot)**

Figure 4 shows the annual cycle of the daily average STGs between ZAC and other AWSs for each month for different temporal resolutions (sample size inscribed in the bar plot). As expected, winter (DJF) is dominated by the strongest inversions, whereas inversions become weaker, and less frequent as the summer (JJA) approaches. The strongest (mean = ~14 °C km⁻¹; 25th–75th percentiles = 7–20 °C km⁻¹) and most frequent (~75 % of 285 days) inversions are present within the ZR in February, especially between ZAC and M3, which have an elevation difference of 377 m (Figs. 3 and 4). Above M3 elevation (e.g., M6), inversions become weaker (6 °C km⁻¹) and less frequent (63 % of 113 days), however, the period used to calculate $STG_{M6-ZAC}$ does not necessarily overlap with the $STG_{M3-ZAC}$ period. Nevertheless, a similar pattern was observed when overlapping days for ZAC, M3, and M6 stations were considered to calculate STGs (for winter 356 overlapping days: $STG_{M3-ZAC}$ = 12 °C km⁻¹, 70 %, and $STG_{M6-ZAC}$ = 5 °C km⁻¹, 61 %). Also, during winter, strong and frequent inversions exist between ZAC and stations on APO (especially AP1; mean $STG_{AP1-ZAC}$ = 8 °C km⁻¹, ~72 % of 802 days), (Figs. 3 and 4).

As spring (MAM) and early summer approach, inversions within the valley gradually become weaker and less frequent (Figs. 3 and 4). During spring, the mean $STG_{M3-ZAC}$ is ~9 °C km⁻¹ (4–14 °C km⁻¹), and the frequency of inversions is 59 % of 1131 spring days. In summer, particularly during the snow-free period, STG becomes steeper, however, mean $STG_{M3-ZAC}$ still shows a positive gradient (1–2 °C km⁻¹) for 45 % of 1440 summer days. In September, STGs are steepest (most negative, -5.0–-3.5 °C km⁻¹), and inversions are rare (1–12 % of days in September); when the valley is completely snow-

free and prevailing positive net radiative balance promotes mixing (Figs. 3 and 4). Hence, the temperature decreases with elevation in the entire valley system.

One of the main factors influencing the $T_{2m}$ via the energy balance is the surface type. The spatial distribution of snow on the ground can impact the vertical temperature gradient (Kirchner et al., 2013) as well as STG. The daily mean $STG_{M3-ZAC}$ when snow is present (mean = 4 °C km$^{-1}$; 25th–75th percentiles = -4–10 °C km$^{-1}$) and absent (-0.3 °C km$^{-1}$; -5–3 °C km$^{-1}$)

at both stations is significantly different (p < 0.01), (Fig. S6). When snow is present at both stations, the daily mean $STG_{M3-ZAC}$ is shallowest and most positive (4 °C km$^{-1}$) compared to $STG_{M3-ZAC}$ when snow is absent either at ZAC (-4 °C km$^{-1}$) or at both stations (-0.3 °C km$^{-1}$). However, the difference was not statistically significant (p > 0.05) between $STG_{M3-ZAC}$ when snow is present at both stations and only at ZAC (2 °C km$^{-1}$), which keeps the valley bottom (surface) cold, favoring inversions. In contrast, $STG_{M3-ZAC}$ is steepest (-4 °C km$^{-1}$) when snow is present only at M3, which causes the temperature to

decrease with elevation. This highlights the importance of snow, which governs the evolution of STG in the ZR.

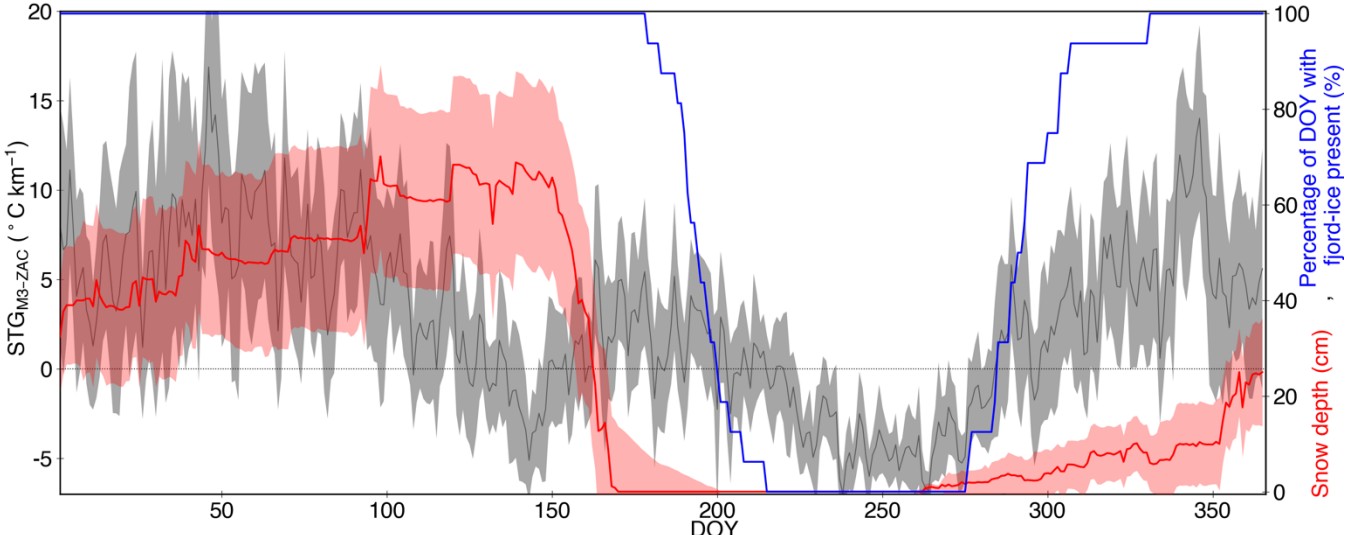

**Figure 5. Annual cycle of STG between ZAC and M3 ($STG_{M3-ZAC}$ in °C km$^{-1}$) and snow depth (in cm) measured at ZAC for 2004–2019. The black and red lines denote the daily mean $STG_{M3-ZAC}$ and snow depth, and the grey and red shaded areas denote a 95 % confidence interval around the given mean, respectively. The blue line represents the percentage of the respective day of the year**

**(DOY) during observation period (2004–2019) with fjord-ice present in Young Sound. The leap year day 29 February is excluded**

Another local factor that can influence the temperature anomaly in the ZR is the fjord-ice break-up and formation period in Young Sound (Figs. 1 and 5). Figure 5 depicts the annual cycle of daily mean $STG_{M3-ZAC}$ and snow depth, and the condition of fjord-ice. Even though the STG is strongly controlled by the surface type (presence and absence of snow), the temporal evolution of fjord-ice can amplify or reduce the prevailing climate anomaly. The shallowest $STG_{M3-ZAC}$ is observed

when Young Sound is completely covered by ice whereas the steepest $STG_{M3-ZAC}$ when the fjord was ice-free. A similar annual relationship was observed between $STG_{AP1-ZAC}$ and the condition of fjord-ice (not shown) implying that the fjord-ice-related anomaly is largely dominated within the surface layer close to ZAC.

## 3.2 Physical controls of STG

To test the hypothesized potential relationship between the STG in the ZR and large-scale drivers, we utilized synthesis of observations (AWSs), reanalysis (ERA5), and RACMO. In particular, the potential relationships between the STG, large-scale atmospheric circulation, surface and atmospheric moisture, and sea ice over the Greenland Sea were examined.

### 3.2.1 Large-scale atmospheric circulation

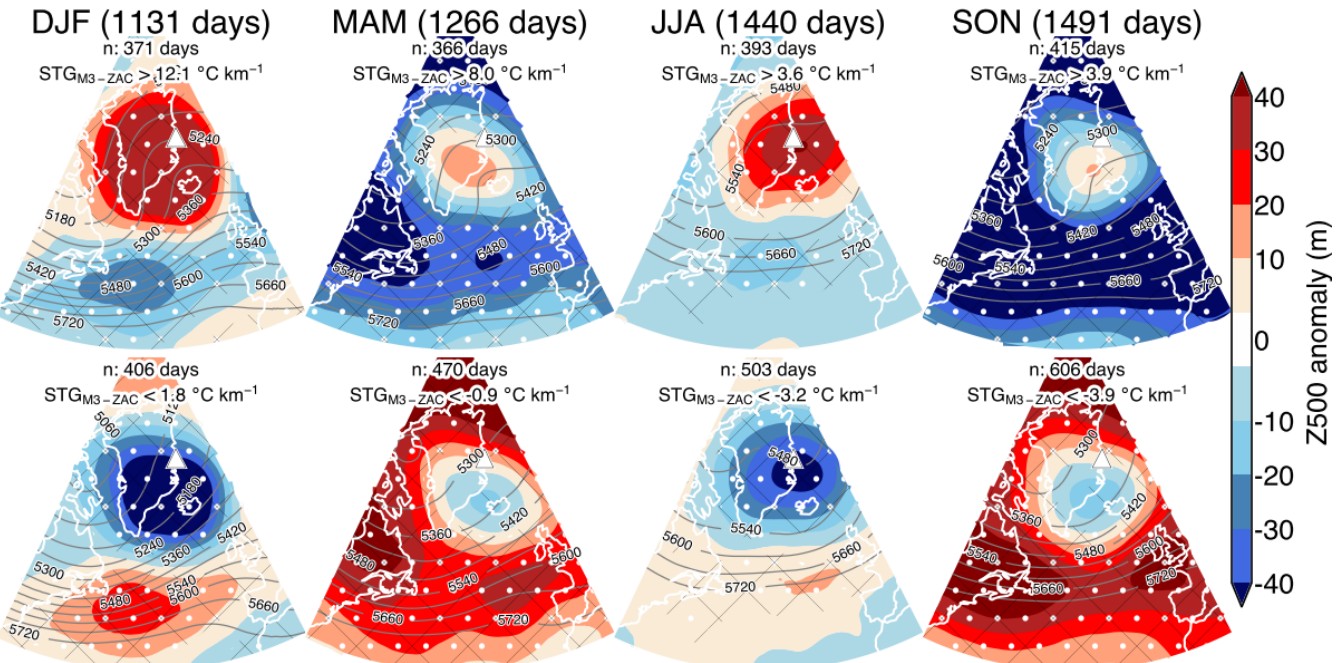

**Figure 6. Composite of geopotential height at 500 hPa (Z500 in m) anomalies (shading) and means (grey contours; 60 m interval) from ERA5 corresponding to the STG between M3 and ZAC (STG$_{M3-ZAC}$) for entire daily measurements (number of days indicated at the top of each seasonal column) for a given season (2004–2019). The upper and lower panels represent Z500 anomaly and mean corresponding to high and low STG days i.e., n number of days when STG$_{M3-ZAC}$ exceed and is less than the indicated STG values in °C km$^{-1}$, respectively, for the given season. The triangle represents the location of the Zackenberg region. The white dots and areas within the black mesh indicate statistically significant differences between high and low composite anomalies at the 0.05 and 0.1 significance levels, respectively**

The role of the large-scale atmospheric circulation in the variation of STG is investigated by a composite analysis. Figure 6 represents composite anomalies of Z500 corresponding to high ($STG_{M3-ZAC} > STG_\mu + 0.5\,STG_\sigma$) and low ($STG_{M3-ZAC} < STG_\mu - 0.5\,STG_\sigma$) STG days between M3 and ZAC (STG$_{M3-ZAC}$) for a given season (2004–2019). Note the similarities between the seasonal anomaly patterns for both the high and low STG days, and the contrast between high and low STG days for a given season. Shallower (steeper) STGs are associated with a positive (negative) anomaly in Z500 relative to the seasonal mean (2004–2019) over eastern Greenland. A similar pattern is observed over the Greenland Sea, but only during winter and summer. An opposite anomaly pattern of Z500 is observed over the southern (equatorward) node of the NAO (where the Azores High resides). Furthermore, the southerly wind pathway is shifted to the northwest (upper-air

ridging) and southeast direction during high and low STG$_{M3-ZAC}$ days, respectively (Fig. 6). The positive (negative) anomaly

of Z500 over the ZR indicates that the upper air (~5500 m) is warmer (colder) than the mean atmospheric state, which leads to shallower (steeper) STG$_{M3-ZAC}$ than the mean STG$_{M3-ZAC}$ (Fig. 6). Hence, along with the surface-bound factors such as snow and topographic shading, and the temperature gradient within the ZR is dependent on synoptic conditions.

When replacing STG$_{M3-ZAC}$ with data from other stations, e.g., STG between M6 and ZAC, and AP1 and ZAC (not shown), etc., the relationship to Z500 anomalies remains unchanged. Also, using all the aforementioned criteria to isolate the

composites result in a similar anomalies pattern (not shown). This implies that the method used in this study can detect a robust statistical linkage between STG and atmospheric circulation patterns.

The anomaly composites of P$_{surf}$ corresponding to high and low STG$_{M3-ZAC}$ days show a similar pattern as Z500, while the centers of both anomalies are shifted eastward (Fig. S7). The shallower (steeper) STG$_{M3-ZAC}$ is associated with higher (lower) P$_{surf}$ compared to the seasonal mean (2004–2019) over the ZR. This implies that STGs are steeper (shallower) in the

case of cyclonic (anticyclonic) circulation patterns. The negative anomaly of P$_{surf}$ does not necessarily imply the cyclonic condition over the ZR (Fig. S7). However, it indicates below average P$_{surf}$, which is also in line with P$_{surf}$ anomaly measured at ZAC for high (0.2–4.1 hPa) and low (-4.5–-0.5 hPa) STG$_{M3-ZAC}$ days. This means that large-scale synoptic patterns are in line with local measurements.

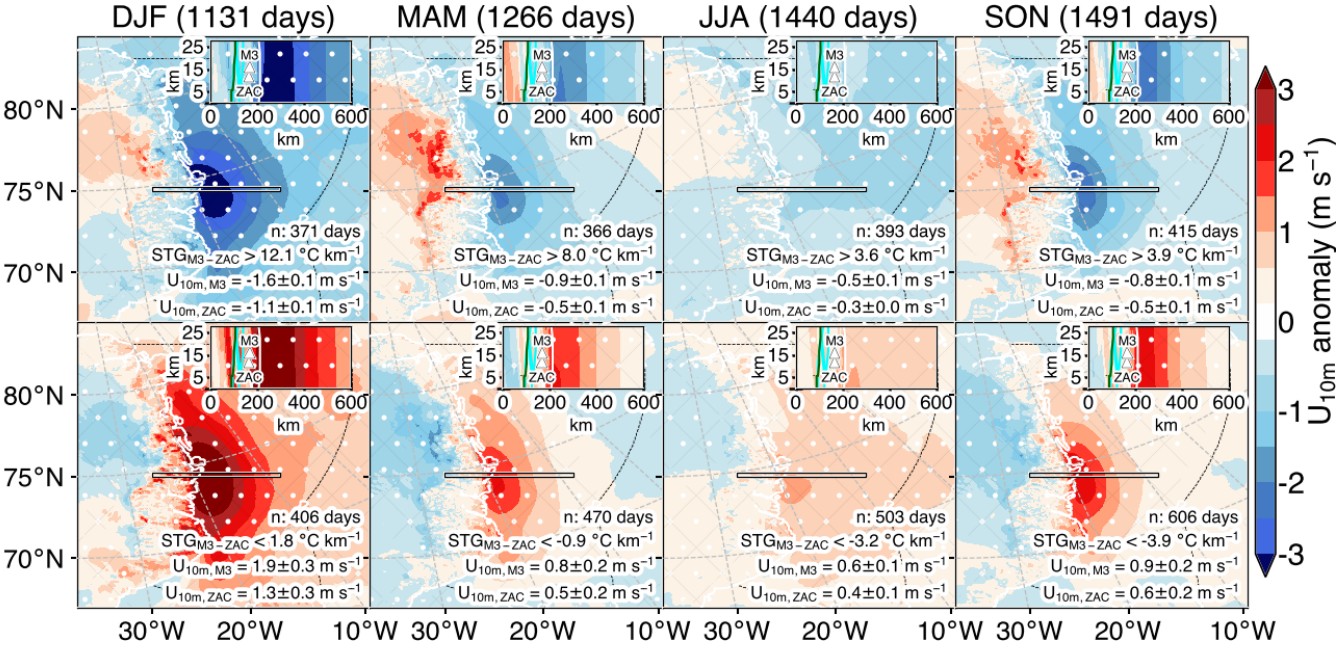

**Figure 7. Idem as Fig. 6, but for the composite of wind speed at 10 m (U$_{10m}$ in m s$^{-1}$) anomalies from RACMO. The inset figure shows the zoomed-in version of the black rectangle in the main figure and Fig. 1 encompassing the ZR; the location of the stations (triangle), the land-sea border (white contour), peripheral glacier (cyan contour), and the GrIS (green contour) are indicated in the inset figure. The dashed black line represents the Greenland Sea coverage. The Greenland Sea is defined by the International Hydrographic Organization (available online at http://www.marineregions.org/). The mean U$_{10m}$ anomaly values (in m s$^{-1}$)**

**interpolated to the station location are also shown in the lower right corner indicating a 95 % confidence interval (calculated using**

As another source of impact, the vertical temperature gradients are controlled by the vertical atmospheric mixing strength and thus a strong relationship between STG and U can be expected (Pepin, 2001). Steeper (shallower) $STG_{M3-ZAC}$
are associated with higher (lower) $U_{10m}$ at both stations (Fig. 7). At both stations (M3 and ZAC), the observed $U_{2m}$ anomaly is similar to the anomaly pattern shown by RACMO using the same reference period (Table S4). Note that the $U_{10m}$ over the Greenland Sea (closest to the study area) consistently shows a similar anomaly pattern as in the ZR (Fig. 7) indicating a similar effect of the U on the STG both over water and coastal region. Findings are independent of station pairs used for characterizing STG (not shown). The strongest inversions are observed for an average $U_{2m}$ of 2–4 m s$^{-1}$ (Fig. S8). However,
as the $U_{2m}$ increases above 4 m s$^{-1}$, the $STG_{M3-ZAC}$ becomes steeper (and inversions are weakened).

### 3.2.2    Near-surface and atmospheric moisture

The shallower $STG_{M3-ZAC}$ is associated with lower $q_{2m}$ over the Greenland Sea compared to the reference seasonal mean over water except for summer (not shown). The computed (from station) and interpolated (from RACMO) $q_{2m}$ anomalies depict a similar pattern with respect to the same baseline period (Table S4).

On average, the inversion strength decreases (less positive or negative) with increasing CCF by enhancing the emission of the $LWR_{in}$ (Nielsen-Englyst et al., 2019). During the high (low) $STG_{M3-ZAC}$ days, the CCF is less (more) in the ZR (Table S4). The computed CCF anomaly is in line with the result from RACMO.

Cloud microphysical composition (such as ice and liquid water cloud content) is radiatively more important than CCF to the near-surface climate. The change in cloud content affects the cloud optical thickness and the integrated precipitation
over Greenland (Noël et al., 2018), and hence, can potentially affect the STG over the ZR. Therefore, we explore the association between the daily IWP and LWP, and STG. We find that during high (low) $STG_{M3-ZAC}$ days both IWP and LWP are smaller (larger) than the baseline average (not shown).

Weather conditions such as precipitation events dampen the inversion strength i.e., steepens the STG, by favoring turbulent mixing through atmospheric convection or orographic lifting. We find that both solid (snow) and liquid (rain)
precipitation from RACMO is higher (lower) during low (high) $STG_{M3-ZAC}$ days (Fig. S9).

### 3.2.3    Fractional sea-ice cover

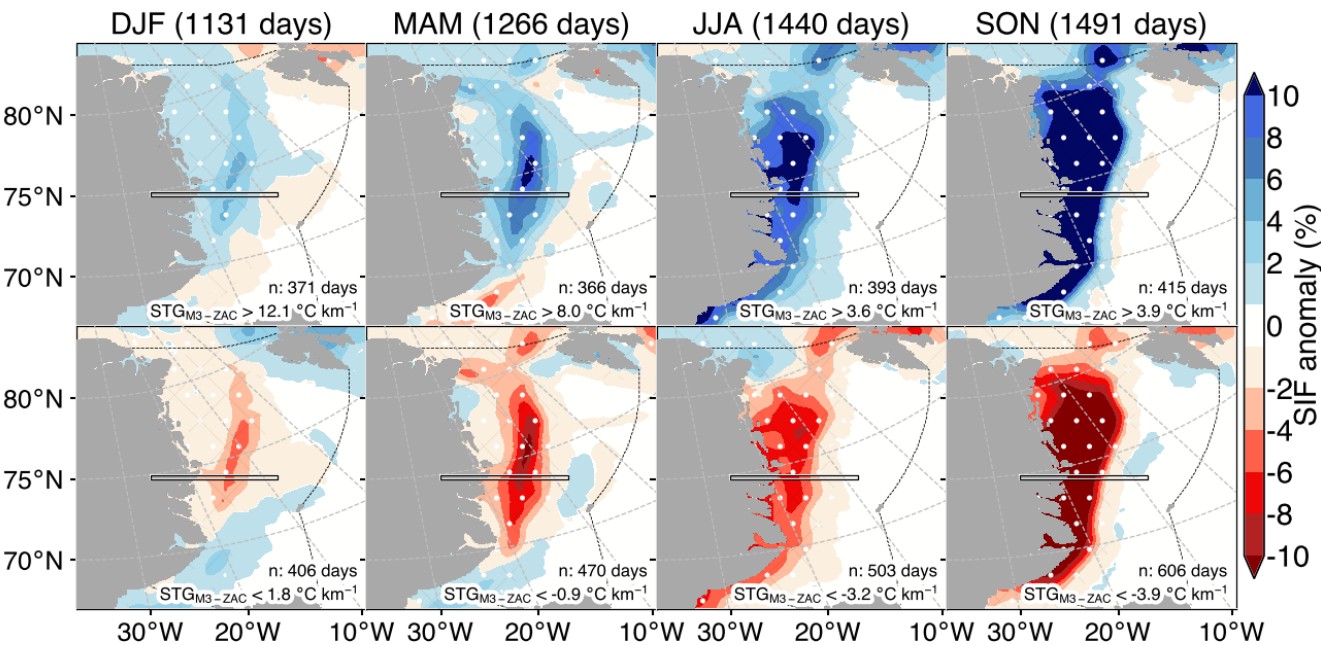

**Figure 8. Idem as Fig. 7, but for the composite of fractional sea-ice cover (SIF in %) anomalies from ERA5 interpolated to the RACMO grid**

Above we show that large-scale patterns relate consistently to local atmospheric stability conditions in the ZR. As a next driver, we assess the variability of sea ice, its export through the Greenland Sea, and its role for local weather conditions in the ZR (Hansen et al., 2008). Figure 8 presents the composites of SIF corresponding to the STG in the ZR. This result implies a link between STG and variability in SIF, suggesting that the shallower (steeper) $STG_{M3-ZAC}$ is associated with higher (lower) SIF in the Greenland Sea (2004–2019). Also, the steepest STG is recorded in September (Fig. 4) which is also

the month of the lowest sea-ice extent (Fig. S4). A similar pattern of SIF composite was observed while using other station pairs (not shown).

### 3.3    Impact of sea-ice variability

### 3.3.1    Near-surface and atmospheric processes

To assess the impact of sea ice on atmospheric conditions, and hence the STG in the ZR, high spatiotemporal resolution

datasets are needed. We separated the atmospheric variables corresponding to high and low SIF days in the Greenland Sea (Fig. S10). The result shows that the greatest variability in the atmospheric variable is close to the area of marginal sea ice in the Greenland Sea (Fig. S10). Composites of atmospheric variables at each grid point of RACMO based on high and low SIF were created. As a result, the first part of the analysis is limited to the regions covered by sea ice to identify signals in atmospheric phenomena due to local changes in sea ice.

Over the Greenland Sea, low SIF is associated with lower than normal $P_{surf}$ (Fig. S11). The open water surface favors more evaporation, decreasing the $P_{surf}$ over the sea. The reduction in the $P_{surf}$ associated with low SIF can enhance the convection over the open water surface and associated moisture flux for precipitation. As a result, low SIF corresponds to increased $q_{2m}$ and precipitation amounts over the sea (Figs. S12 and S13). Furthermore, the low $U_{10m}$ coincides with the high SIF over the Greenland Sea (Fig. S14). This result is in line with Jakobson et al. (2019) who found a negative correlation

between SIF and U over the Arctic Ocean (particularly in autumn, winter, and spring) which reflects the reduction in atmospheric stratification and aerodynamic surface roughness associated with a decrease in sea ice. Note that similar patterns for $P_{surf}$, precipitation, and $U_{10m}$ were observed over the Greenland Sea for the high and low STG days in the ZR (Figs. S7, S9, and 7 versus Figs. S11, S13, and S14), which supports the fact that sea ice plays a strong role in influencing lapse rates by controlling the atmospheric conditions in the valley.

**Table 2. Mean anomalies of 2 m air temperature ($T_{2m}$ in ºC) and the slope temperature gradient ($STG_{M3-ZAC}$ in ºC km$^{-1}$) between M3 ($T_{2m, M3}$) and ZAC ($T_{2m, ZAC}$) pair corresponding to high and low fractional sea-ice cover (SIF in %) from ERA5 averaged over the Greenland Sea for the given season (2004–2019). $T_{2m, M3}$, $T_{2m, ZAC}$, $STG_{M3-ZAC}$, and SIF anomalies are calculated using the same period i.e., when the variable is recorded by both AWSs**

| Seasons | DJF | | MAM | | JJA | | SON | |
|---|---|---|---|---|---|---|---|---|
| Composites | High | Low | High | Low | High | Low | High | Low |
| Total days | 1131 | | 1266 | | 1440 | | 1491 | |
| SIF (days) | 41 (57) | -26 (76) | 35 (64) | -36 (64) | 33 (72) | -6 (908) | 34 (75) | -5 (1043) |
| $T_{2m, M3}$ | -2.2 | 3.0 | -3.9 | 5.6 | -1.1 | 0.4 | -5.8 | 2.4 |
| $T_{2m, ZAC}$ | -3.5 | 4.3 | -6.1 | 7.3 | -2.9 | 0.9 | -7.4 | 3.2 |
| $STG_{M3-ZAC}$ | 3.6 | -3.6 | 5.8 | -4.4 | 4.7 | -1.2 | 4.3 | -2.2 |

In the second part of the analysis, we again divided STG and atmospheric variables according to high and low SIF

zonally averaged over the Greenland Sea (blue shaded area in Fig. 1), (Fig. S4). This was done to understand how the change in sea ice over the Greenland Sea on average affects the atmospheric conditions in the near-coastal (terrestrial) area, in particular the ZR. The results show that during high (low) SIF days over the Greenland Sea, the STG in the ZR is shallower (steeper) than the mean values (Table 2); the mean $STG_{M3-ZAC}$ varies by ~4 ºC km$^{-1}$ for a corresponding ~27 % change in SIF over the Greenland Sea. Both stations show a negative (positive) anomaly of $T_{2m}$ corresponding to the high (low) SIF days

(Table 2). However, on high SIF days, the lower (e.g., ZAC) station shows more negative anomaly (~-5 ºC) compared to the higher station (e.g., M3; ~-3 ºC), shallowing the STG; the opposite is true for the low SIF days, thus, steepening the STG.

Other station pairs also showed similar results, e.g., during high SIF days (24–34 % SIF anomaly) the $STG_{M6-ZAC}$ is more than the mean value (1.5–2.7 ºC km$^{-1}$), while during low SIF days (-33–-6 % SIF anomaly) the $STG_{M6-ZAC}$ is relatively less (-1.7–-0.6 ºC km$^{-1}$) than the mean (Table S5).

The temperature difference ($\Delta T$) between the outer (DAN's location) and inner coast (ZAC's location) is also impacted by the change in sea ice. The aerial distance between the outer coast and stations increases in the following order: DAN (2 km) < ZAC (27 km) < M7 (32 km), (Fig. 1). Generally, along the Young Sound (fjord), a strong temperature (and pressure) gradient develops which generates onshore winds (especially in summer, Fig. S3); in winter the north-south pressure difference favors the mesoscale wind field (northerly wind) and will be part of the shallow katabatic outflow from the ice

sheet. In winter, the outer part of the fjord is warmer than the inner coast ($\Delta T_{DAN-ZAC}$, ~1 ºC for 1996–2015), which indicates the maritime influence due to the existence of partially open water at the opening of Young Sound (Hansen et al., 2008). Summertime, however, is characterized by an opposite temperature gradient ($\Delta T_{DAN-ZAC}$ ~-0.7 ºC for 1996–2015); the cold Greenland Sea and drifting sea ice make areas near the ocean colder than snow-free inland areas. The continentality increases rapidly towards inland ($\Delta T_{DAN-M7}$ ~-3–-1 ºC for 2009–2015). Interestingly, $\Delta T_{DAN-ZAC}$ shows a consistent anomaly

during the high and low SIF days for all seasons; the $\Delta T_{DAN-ZAC}$ is higher (lower) than the mean for high (low) SIF days (Table S5). This implies that areas near the coast (DAN) are warmer (colder) than inland (ZAC) during high (low) SIF days. The reason for the observed anomalous pattern in continentality might be due to warmer and fresher surface water being trapped in the inner part of the fjord due to a high SIF (or multi-year sea ice), as reported by Stranne et al. (2021) in a fjord in North Greenland and possibly increasing the local temperature. Moreover, the sea-ice damming effect might influence the

outer part of the fjord more than the inner part, causing a positive (negative) coast-inland temperature gradient anomaly during high (low) SIF days.

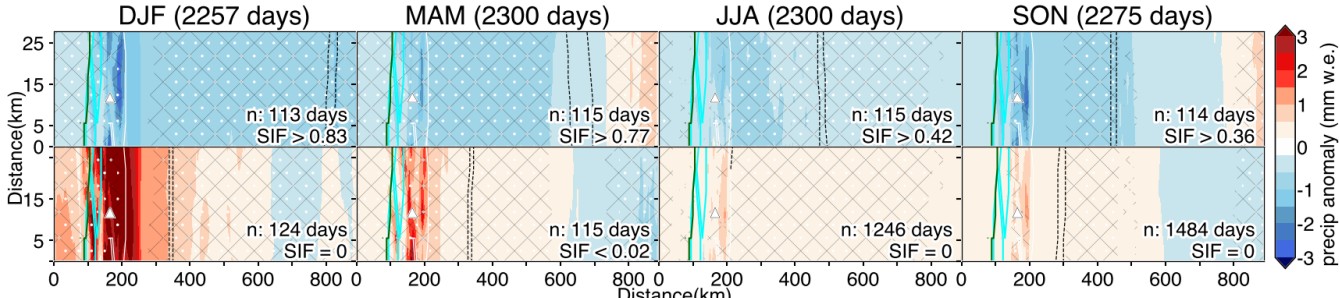

**Figure 9. Composite of precipitation (precip in mm w.e.) anomalies from RACMO corresponding to zonally (overall grid cells) averaged (median) SIF over the Greenland Sea for entire daily measurements for the given season (1996–2020). The upper and**
**lower panels represent precipitation anomaly corresponding to high and low SIF days i.e., n number of days when SIF exceed and is less than or equal to the indicated SIF values, respectively, for each season. The figure is the zoomed-in version of the black rectangle in Fig. 1 encompassing the ZR; the location of the stations (triangle), the land-sea border (white contour), peripheral glacier (cyan contour), and the GrIS (green contour) are indicated. The average sea-ice extent (dashed black line) for the given condition is also shown; the outer line represents 0.25 SIF and the inner line 0.3. The white dots and areas within the black mesh**
**indicate statistically significant differences between high and low composite anomalies at the 0.05 and 0.1 significance levels, respectively**

       The change in the sea ice has a similar impact on the terrestrial climate of the ZR as it has over the Greenland Sea. On high SIF days, the sea ice extends ~600 km east of Zackenberg while on low SIF days, it extends ~100 km east (Fig. 9) on average. This can have several implications on Zackenberg's climate. Over the ZR, the $T_{2m}$ anomaly composite from

RACMO (Fig. S15) shows a similar pattern as observed by the climate station (Table 2) for days with a high and low SIF. Also, both calculated and interpolated $q_{2m}$ anomaly composites show a consistent pattern for high and low SIF days (Table S6). The enhanced atmospheric instability (convection), availability of moisture, and increase in $U_{10m}$ associated with a reduction in sea ice and its extent bring more precipitation to the ZR (Fig. 9), and consequently, the STG becomes steeper.

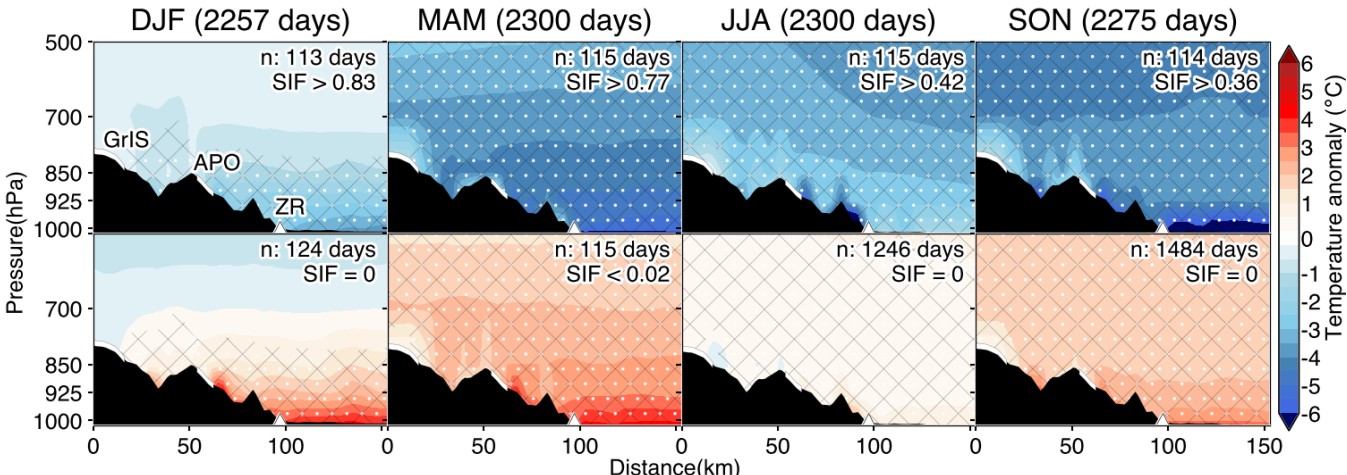

**Figure 10. Idem as Fig. 9, but for the composite of the cross-sectional transect (XY as shown in Fig. 1) of vertical air temperature (T in °C) anomalies profile. The portions of the Greenland Ice Sheet (GrIS), A. P. Olsen Ice Cap (APO), and the Zackenberg region (ZR) passing through the transect are labelled in the upper panel of DJF season.**

To understand how the sea-ice variability affects the vertical temperature gradient, we further divided the vertical air temperature along the XY transect shown in Fig. 1 (Fig. 10). Evidently, the sea-ice variability causes a nonlinear vertical
response; it has a stronger influence in the low-level atmosphere compared to the upper level. Mostly, the negative (positive) anomaly of temperature is associated with high (low) SIF days. However, the temperature anomaly is more negative (positive) in the low-level atmosphere close to sea ice compared to the upper level for high (low) SIF days. On average, when SIF increases by 48 % (39–56 %), the temperature decreases by -4 °C (-7– -2 °C) at 1000 hPa and -3 °C (-4– -0.3 °C) at 500 hPa along the transect, and when SIF decreases by 37 %, the temperature increases by 4 °C at 1000 hPa and 0.7 °C at
500 hPa along the transect (Fig. 10). This result suggests that the variability of sea ice has a greater impact on the air close to the surface (at 1000 hPa) than higher above (at 500 hPa).

### 3.3.2 Surface mass balance of a local ice cap

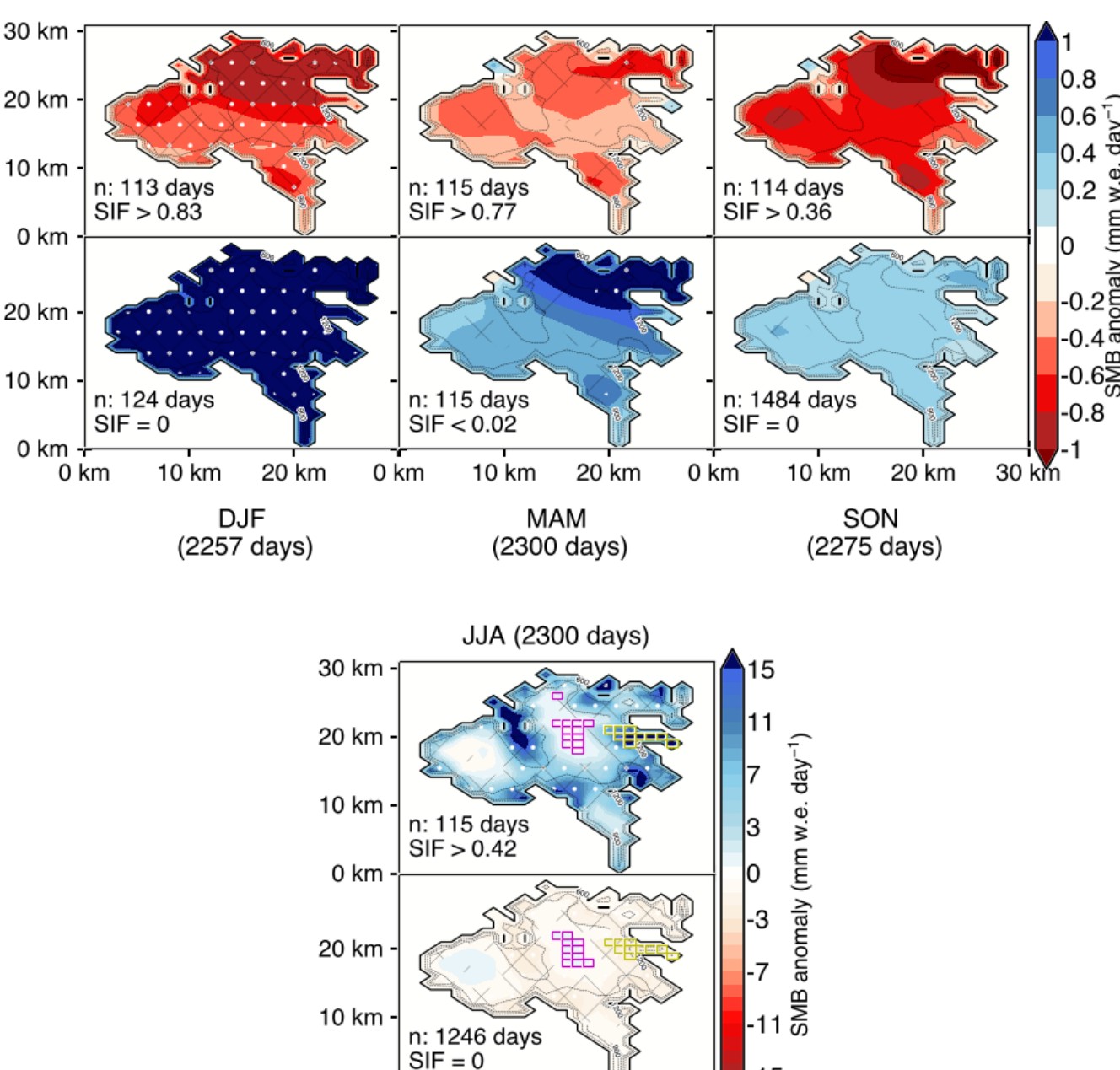

**Figure 11. Idem as Fig. 9, but for the composite of daily surface mass balance (SMB in mm w.e. day$^{-1}$) anomalies of A.P. Olsen Ice Cap (APO), (location in Fig. 1) from RACMO. In JJA panel, the magenta and yellow rectangles indicate the grid cells whose values are used for the calculation of the average SMB anomaly on accumulation (>1100 m a.s.l.) and ablation (<1100m a.s.l.; only the southeastward flowing outlet glacier) area, respectively, of the APO. Note the use of a separate color bar for summer composite due to large mass overturn**


An interesting question related to the STGs and their connection to the sea ice is the effect on the energy or mass balance of the local glaciers and ice caps. We examined the SMB of the APO from RACMO to assess the relevance of the STG and SIF variability for estimating the SMB of the glacier. Our main focus is the southeastward flowing outlet glacier as long-term atmospheric and glaciological measurements exist there (Citterio et al., 2017). We used an equilibrium line altitude of 1100 m a.s.l. following Noël et al. (2019) to estimate a typical separation between accumulation and ablation areas of the APO. For all seasons except summer, the daily mean SMB is less (more) than the reference SMB (1996–2020), (Fig. S16) during high (low) SIF days (Fig. 11). This is true for both the accumulation and ablation area of the APO. For high SIF days, the reduction of daily SMB (-0.8–-0.5 mm w.e. day$^{-1}$) can be attributed to the reduction in precipitation and $q_{2m}$, leading to less accumulation of snow (Fig. S17); conversely, the increase in precipitation (in the form of snow) and $q_{2m}$ associated with low SIF could increase SMB (0.3–2.2 mm w.e. day$^{-1}$). Furthermore, despite $T_{2m}$ being higher than the reference during days with low SIF, $T_{2m}$ remains largely negative (Fig. S18) and does not lead to increased melt, but increased snowfall rates (Schweiger et al., 2008).

The summer SMB anomaly pattern is different from the rest of the season in that anomalies show a different signal in the accumulation and ablation areas (Fig. 11). Overall, the SMB anomaly becomes more negative (positive) with elevation on high (low) SIF days. In the ablation area of the APO, the positive anomaly of SMB (~16 mm w.e. day$^{-1}$) is related to high SIF days, indicating less melt than on average. In contrast, for low SIF days, the SMB over the ablation area shows a negative anomaly (~-3 mm w.e. day$^{-1}$), indicating more melting. The decrease (increase) in $T_{2m}$ associated with high (low) SIF days, which reduces (enhances) the melt rates, can possibly explain the resultant positive (negative) SMB anomaly in the ablation area; for high (low) SIF days $T_{2m}$ anomaly is -2 (~0.01) ºC. Also, the positive anomaly of the $LWR_{in}$ (related to an increase in atmospheric moisture and liquid-bearing cloud) (not shown) associated with low SIF days can further explain the negative SMB anomaly.

In the accumulation area of the APO, the summer SMB anomaly pattern shows a vertical gradient; in high (low) SIF days, the SMB anomaly changes from positive (negative) to negative (positive) value with elevation. On average, the SMB anomaly is positive (negative) for high (low) SIF days in the accumulation area. However, the anomaly sign changes with elevation. In higher reaches of the accumulation area (above ~1317 m a.s.l), for high SIF days the daily mean SMB is less than the reference SMB (~-0.3 mm w.e. day$^{-1}$); for low SIF days, the SMB anomaly is ~0.06 w.e. day$^{-1}$ (above ~1157 m a.s.l), (Fig. 11). The decrease (increase) in precipitation associated with high (low) SIF days leads to less (more) accumulation of snow, and hence less (more) SMB with respect to the baseline mean.

### 3.3.3    On the temporal robustness of the relationship between sea-ice evolution and near-surface atmospheric conditions

To account for the impact of the multi-annual evolution of sea ice on near-surface atmospheric conditions, we further derived the composites of near-surface and atmospheric variables for the subperiods of the SIF time series defined by the detected change points (see Sect. 2.9.2). All the previously identified near-surface, atmospheric, and the SMB (of the APO)

anomaly patterns for high/low SIF remain consistent for the subperiods independent of the change in the multi-annual mean (shown in Table S7 for STG$_{\text{M3-ZAC}}$, Figs. S15 and S19 for T$_{2m}$ and APO SMB, respectively). On average, relative to the early period (1996–2001), the median daily SIF during the high SIF day is 24 % (15–39 %) less in the recent period (2014–2020); with the largest reduction in winter (39 %; Fig. S5).

Within all the subperiods, on average, the STG$_{\text{M3-ZAC}}$ varies by 3–5 ºC km$^{-1}$ for a corresponding ~21–28 % change in SIF over the Greenland Sea. During the high SIF days, the difference in mean daily STG$_{\text{M3-ZAC}}$ in the recent period (2014–2019) with respect to the early period (2004–2013) shows opposite signs for winter and summer seasons. During winter, the temperature inversion strength decreased in the recent period relative to the early period (by 2.5 ºC km$^{-1}$), whereas in summer the temperature inversion strength increased (by 1.2 ºC km$^{-1}$). This seasonal contrasting change is in line with the inversion trend (1979–2017) pattern found by Shahi et al. (2020) in Northeast Greenland.

The mean atmospheric condition at the ZR also varies with the temporal change in sea ice, which is well reflected by the respective RACMO climate anomaly patterns (Fig. S15). For high SIF, the mean daily T$_{2m}$, precipitation, and q$_{2m}$ is higher in the recent period (2014–2020) compared to the early period (1996–2001), where the mean daily T$_{2m}$, precipitation, and q$_{2m}$ increased by 4 °C (1–7 °C), 1 mm w.e. (0.3–2.1 mm w.e.) and 0.5 g kg$^{-1}$ (0.2–0.7 g kg$^{-1}$), respectively. At the same time, the air temperature of the vertical profile (1000–500 hPa) along the XY transect increased as well (corresponding to a 24 % decrease in median SIF). The magnitude of the increase is larger close to the surface (~3 °C at 1000 hPa) than in the upper atmosphere (~2 °C at 500hPa).

It is important to note that the daily SMB anomaly pattern remains consistent with the time evolution of sea ice (Figs. 11 and S19), however, the absolute magnitude of the anomaly differs. In the summer of the recent period (2008–2020), high SIF is associated with the daily mean SMB anomaly being higher in both the accumulation (by 2 mm w.e. day$^{-1}$) and the ablation (by 6 mm w.e. day$^{-1}$) areas of the APO compared to the early period (1996–2001). The resultant increase in the daily SMB anomaly is rather probably because of the weather condition associated with decreasing SIF (by 14 %), e.g., the daily T$_{2m}$ anomaly increased by 2 °C and the snowfall anomaly decreased by -1.3 mm w.e. day$^{-1}$.

The daily mean SMB also varies with time and is related to the SIF. During the high SIF days, in the recent period (2008–2020), the daily mean summer SMB of APO decreased in both the ablation (-2.5 mm w.e day$^{-1}$) and accumulation (-0.2 mm w.e. day$^{-1}$) areas relative to the early period (1996–2001). However, the increase in the daily mean T$_{2m}$ (by ~4 °C) and decrease in snowfall (by ~2 mm w.e. day$^{-1}$) associated with the multi-annual decreasing SIF (by 14 %) might explain the observed general reduction in the SMB of APO during the high SIF days. In contrast, in all seasons except summer during the high SIF days, the daily mean SMB is higher in the recent period compared to the early period (~0.2 mm w.e. day$^{-1}$). Apparently, the increase in daily mean snowfall in the recent period (~0.4 mm w.e. day$^{-1}$) compensates for the increase in daily mean T$_{2m}$ (2.6 °C), above the APO ice cap might explain the increase in the daily mean SMB. It is important to keep in mind that despite the temperature increased in the recent period, it is still below the freezing point (~-16 °C), hence does not contribute to melting.

 **4    Discussion**

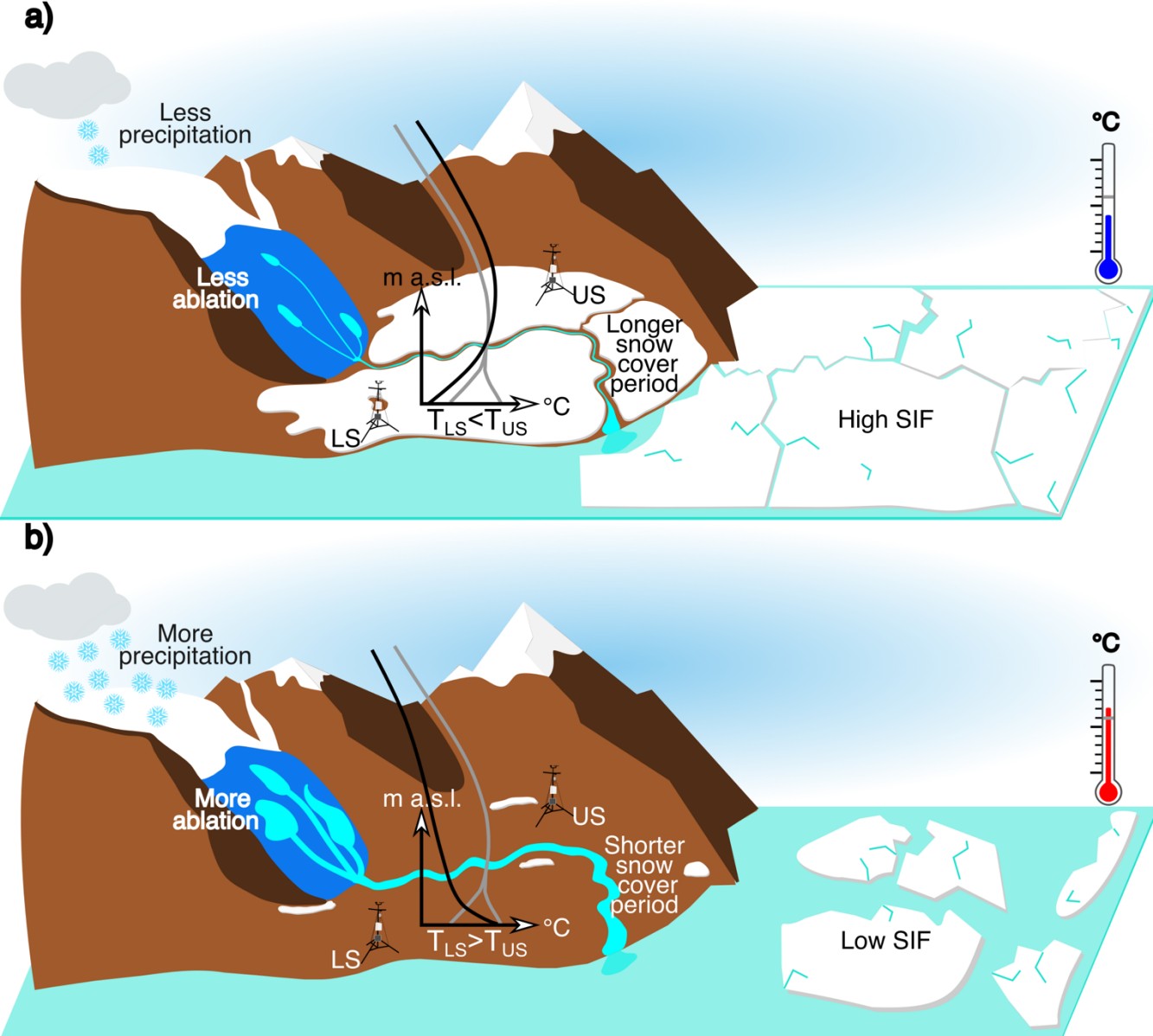

**Figure 12. A schematic representation of some important linkages in the Zackenberg region (ZR), fractional sea-ice cover (SIF) in the Greenland Sea, and surface condition of A. P. Olsen Ice Cap (APO) especially for the summer season. The temperature at the lower station (LS), $T_{LS}$, can be higher or lower than the temperature at the upper station (US), $T_{US}$. (a) The upper panel shows the conditions when $STG_{US-LS}$ is shallow ($T_{LS}$ is less than $T_{US}$; less negative or positive (inversion) $STG_{US-LS}$), when there is more snow in the valley, when SIF is high, along with low precipitation amounts, and little snow/ice ablation at APO. (b) The lower panel shows the opposite conditions when $STG_{US-LS}$ is steep ($T_{LS}$ is more than $T_{US}$; more negative or less positive $STG_{US-LS}$), when there is less snow in the valley, when SIF is low, along with high precipitation amounts, and strong snow/ice ablation at APO. The solid grey line represents a climatological mean of the temperature profile in the ZR and the corresponding solid black lines represent the mean temperature profile in each condition**

The STG in the ZR shows a pronounced seasonal cycle, with the strongest inversions occurring mostly in winter (Fig. 4). The formation of temperature inversions in the ZR is mainly due to radiative cooling of the snow surface (Fig. 12), particularly during polar night (Hansen et al., 2008; Shahi et al., 2020). Moreover, ZAC shows the lowest mean winter $T_{2m}$
(~-19 ºC) compared to stations higher up for all overlapping periods (Fig. 3). Thus, the ZAC location (valley bottom) might represent the approximate elevation of the inversion base (lowest elevation of the inversion layer), especially during winter. The occurrence of sporadic rain and breeze, which cause mixing of the lower atmosphere can dissolve the inversions (Fig. S1b).

During winter, strong and frequent positive STG between ZAC and stations on APO can be attributed to cold air
pooling in the valley, resulting in temperature inversions within the valley, whereas the APO remains above the inversion layer. Clearly, this information is imperative in the glacier mass balance modelling perspective for simulating temperature on the glacier back in time using the longest recorded information from ZAC.

Spring and (early) summertime inversions in low-lying coastal regions like the ZR can be attributed to the cooling of the atmospheric layers above the surface due to the consumption of energy for snowmelt (Serreze and Barry, 2014). During
summer, STG on the valley floor might be influenced by low-level marine stratus clouds brought by the prevailing sea breeze from the outer coast (Hansen et al., 2008), (Fig. S3). Furthermore, the base of low clouds (fog) on the mountain slopes can also increase due to the mixing within the lower atmosphere, which can lift the inversion base from 300 to 600 m by advection and create elevated inversions (inversions whose base is above the surface), (Hansen et al., 2008).

The Z500 composites we use to attribute days with high and low STG do not necessarily correspond to GBI anomalies
since a large area is averaged (60–80° N, 20–80° W region) to derive the latter. We elaborate on the variations of STG at the local scale in response to the spatial variability of composites anomaly of Z500. The positive (negative) anomaly of Z500 strengthens (weakens) the Greenland anticyclone, which increases (decreases) the downward vertical motion of air, consequently leading to positive (negative) air temperature anomalies. Locally, we find that this effect is stronger in higher elevations (e.g., M3, M6) compared to lower elevations (e.g., ZAC), which results in shallower (steeper) STG in response to
the positive (negative) anomaly of Z500 and $P_{surf}$ over the ZR. In particular, subsidence of air in a high-pressure system can cause cold and dry air to drain along the valley slopes, and hence, favor the development of inversions. We find that inversion frequency and strength can be related to large-scale atmospheric conditions.

The variability in the export of sea ice through the Fram Strait and in the Greenland Sea has a regional-scale influence on the atmospheric condition along the east coast of Greenland (Hansen et al., 2008). In recent years, declining sea ice
enhances heat and moisture transfer between the ocean and the atmosphere, resulting in an amplified Arctic warming (Serreze et al., 2009; Screen and Simmonds, 2010). We show that the atmospheric conditions in the Greenland Sea and locally in the ZR, such as $P_{surf}$, $T_{2m}$, U, CCF, and precipitation, are influenced by SIF conditions (Fig. 12). A comparison between the zonally averaged SIF over the Greenland Sea and $T_{2m}$ and precipitation in the ZR indicates that seasons with a SIF below (above) the reference period (1996–2020) are warmer (colder) and wetter (drier) than the average. Additionally,

an increase (decrease) in $q_{2m}$, CCF, and U is associated with low (high) SIF days. These results are in line with other studies that show a strong linkage between sea-ice reduction and atmospheric warming in surrounding areas (Bhatt et al., 2010; Comiso, 2002; Hanna et al., 2004; Serreze et al., 2011; Stroeve et al., 2017), and increasing U, tropospheric moisture, CCF, and precipitation in the Arctic Seas (Stroeve et al., 2011; Müller et al., 2021; Deser et al., 2000; Francis et al., 2009; Jakobson et al., 2019; Overland and Wang, 2010; Schweiger et al., 2008; Alley et al., 2006; Schuster et al., 2021).

The STG in the ZR and changes in the sea ice in the Greenland Sea are simultaneous (Fig.12). The atmospheric response to the sea-ice variability can explain large parts of the observed STG variability in the ZR. Low $P_{surf}$, high U and $q_{2m}$, and large snowfall are associated with low SIF anomalies, thus favoring the formation of steep STG. Hinkler (2005) showed that the reductions in sea ice lead to an increase in snowfall, which is in line with our result. Rogers et al. (2005) demonstrated that in mild winters with little sea-ice export through the Fram Strait, stronger and more frequent cyclones pass Northeast Greenland towards the Fram Strait. In contrast, during cold winters with high sea-ice export, cyclonic activity is more prevalent towards the Barents Sea and Eurasia side of the Arctic Ocean. Müller et al. (2021) found a linkage between the likelihood of the occurrence of extreme precipitation over Northwest Svalbard and the reducing sea-ice extent east of Greenland. The atmospheric conditions such as high U, increase CCF accompanying cyclone and precipitation events decrease the stability of the atmosphere, and hence steepen the STG.

Changes in the sea ice could influence the SMB of the GrIS by the change in the near-surface air temperature and moisture (Noël et al., 2014; Overland et al., 2012). The changes in sea ice may also affect the atmospheric conditions above Greenland, thereby affecting the general circulation (Cho et al., 2022). Noël et al. (2014) found that GrIS SMB is insensitive to changes in oceanic forcing because the katabatic winds flowing down the GrIS prevent the oceanic near-surface air from entering inland. In contrast, Stroeve et al. (2017) found a correlation between the timing of melt onset over sea ice within the Baffin Bay and the Davis Strait, and at the GrIS. They showed that an increase in the transfer of turbulent heat fluxes from the ocean to the atmosphere during years with early sea-ice melt leads to higher air temperatures, humidity, and melt over the GrIS. Furthermore, the increase in atmospheric moisture during low sea-ice conditions can also increase surface melt through increased $LWR_{in}$ (Bennartz et al., 2013; Van Tricht et al., 2016).

In summer, the negative SMB anomaly of the APO during the low SIF days is in line with the findings of Stroeve et al., (2017). As a result of reduced sea ice, onshore winds might bring higher temperatures and moisture to the lower part of the APO, which in turn could make the STG more negative (e.g., the average $STG_{AP2-AP1}$ anomaly is -1.5 ºC $km^{-1}$ for ~17 SIF reduction, implying AP1 is warmer than AP2), (Table S5) and increases surface melting there. However, unlike Stroeve et al. (2017), we did not account for the time lag (1-week lag) effect of the heat transfer from reduced sea-ice areas further inland to the GrIS. This might influence the magnitude of the SMB anomaly. However, considering the short distance between APO and water bodies i.e., Young Sound (~35 km) and the outer coast (~75 km), we speculate a shorter time lag between sea ice melt and ice cap response (though elevation might intervene the interaction). Furthermore, the katabatic forcing of the APO could reduce the impact of the near-surface air influenced by the ocean on the glacier boundary layer.

In Northeast Greenland's peripheral glaciers, Khan et al. (2022) observed a recent (October 2018–December 2021) increase in snowfall at high altitudes partially offset the recent increase in the melt at low elevations. According to the authors, the glacier plateau's high elevation (2000–3000 m) likely plays a crucial role in maintaining a viable accumulation area. This is consistent with our observed contrasting signals of the summer SMB anomaly of the APO in the accumulation and ablation areas, especially during low SIF days. Thus, in addition to plateau geometry, decreasing sea ice likely leads to the observed contrasting patterns in the accumulation and ablation area of Northeast Greenland's peripheral glaciers, especially during the period 2018–2020 as shown by Khan et al. (2022), (Fig. S4).

One of the manifestations of climate change is a decline in the sea ice (Stroeve and Notz, 2018; Peng and Meier, 2018; Polyakov et al., 2022). Our study shows that local change in the sea ice due to changing climate impacts the ZR atmospheric conditions. However, even with the observed decline in Arctic sea ice, the atmospheric anomaly with respect to high and low SIF remains consistent. Probably with climate change the atmospheric conditions governed by less sea ice (low SIF days) will be more prominent in the future. While we show that lower sea ice coincides with a decrease in atmospheric stability, we hypothesize a non-linear response on SMB or runoff, which makes predictions out of the scope. In general, decreased stability and increased moisture transport point towards increased precipitation and hence an accelerated water cycle. This is in line with projections of Arctic climate change and should be studied in detail on a regional scale both using observations and model results (McCrystall et al., 2021).

We acknowledge that parts of the connections shown in Figure 12 may indeed also apply in reverse order or are driven by the same large-scale conditions, e.g., higher air temperatures and low sea ice conditions are intrinsically related. While local-scale processes influence the melt variability, the large-scale atmospheric circulation (500 hPa geopotential height anomalies) can be a superimposed driver in controlling both the sea ice and ice cap melt (Stroeve et al., 2017). However, the clear association between sea ice anomalies and nearby coastal climate anomalies indicates a local response.

## 5    Conclusion

Air temperature and its variability in space and time is a key variable of climate change, as a link can be directly established for it with many climate impacts, such as ecosystem or cryosphere changes. This motivated our study on the spatiotemporal variability of temperature gradients in the orographically complex terrain of ZR where an extensive monitoring and analysis program of ecological and cryospheric changes has been established. To step towards a better understanding of underlying processes forcing the near-surface temperature variability in the ZR, this study investigated in particular 2-m air temperature, using the dense network of AWSs and combining data from the UAV, atmospheric reanalysis (ERA5) and a regional climate model (RACMO). More specifically, this study documented an association between the slope temperature gradient, sea ice, and other atmospheric variables in and around the ZR. In addition, we discuss the implications for the SMB of a local ice cap (APO). The main findings are as follows:

– The surface type and fjord-ice conditions are the dominating drivers governing the temporal evolution of the near-surface temperature distribution in the ZR. A snow cover in the valley and an ice-covered fjord favor the formation of shallow STG, possibly inversions, whereas a snow-free valley and an ice-free fjord favor steep STGs.

– The STG within the ZR is strongly affected by the prevailing synoptic scale atmospheric pressure patterns. Shallower (steeper) STGs are associated with a positive (negative) anomaly in Z500 and surface pressure over eastern Greenland. This indicates, when the upper air (~5500 m) is warmer (colder) than the mean atmospheric state, the STG within the valley is shallower (steeper). The patterns of these changes are consistent when different station pairs were employed.

– The change in the sea ice has a similar impact on the terrestrial climate of the ZR and the climate over the Greenland Sea. During high (low) SIF over the Greenland Sea, the STG in the ZR is shallower (steeper) than the mean value, e.g., the mean STG$_{M3-ZAC}$ varies by ~4 ºC km$^{-1}$ for a corresponding ~27 % change in SIF over the Greenland Sea. When sea ice increases, the temperature at the bottom of the valley decreases more than at the top, resulting in a shallow STG. Furthermore, the atmospheric response to the sea-ice variability can explain large parts of the observed STG variability in the ZR. Low surface pressure, high wind speed and specific humidity, and large snowfall are associated with low SIF anomalies, thus favoring the formation of steep STG.

– The change in sea ice also shows an association with SMB change. For all seasons except for summer, the SMB anomaly of the APO is negative (positive) on high (low) SIF days which can be attributed to less (more) snowfall and thus less accumulation. However, it is important to note that the summer SMB anomaly shows a different pattern in the ablation and accumulation area of the APO. During summer, days with high SIF are associated with a positive SMB anomaly in the ablation area (~16 mm w.e. day$^{-1}$) and a negative anomaly in the accumulation area (~-0.3 mm w.e. day$^{-1}$), indicating both less melt in the ablation area and less accumulation in the accumulation area. The decrease in T$_{2m}$ and snowfall related to high SIF days can explain this opposite pattern in the ablation and accumulation area.

This study shows complex relationships between temperatures in low and high elevations that vary with changes in local- and large-scale conditions. Studies that calibrate vertical dependences of environmental variables based on low-elevation measurements and standard methods for spatial extrapolation may miss out on parts of the complexity and hence limit applicability. Additionally, the information on the magnitude and extent of effects from local-scale processes is imperative for an efficient evaluation of regional as well as global climate models. The surface-atmosphere linkages described in this study will also be relevant in other parts of Greenland with a similar topographic setting. Clearly, more process-based studies are needed in order to increase confidence in the cause-effect mechanisms and quantify relationships. Besides, this study highlights the complexity of the intertwined processes like local scale sea-ice variability and terrestrial climate anomaly. We speculate that with climate change the ZR atmospheric conditions like a decrease in atmospheric stability, increase in precipitation, and reduction in SMB of the nearby glaciers and ice caps (especially in summer) governed by declining sea ice will be more prominent in the future. This local-scale change agrees with projections of Arctic climate

change and sets the stage for more detailed process-oriented studies about the effects of climate change on a high Arctic ecosystem.

*Data availability.* Derived data supporting the finding of this study are available from the corresponding author on request.

*Author Contributions.* SS, JA, and WS contributed to the study conception and design. SS downloaded and analyzed the data. The first draft of the manuscript was written by SS and all authors discussed the results and contributed to the final manuscript.

*Competing interest.* The authors declare no conflict of interest.

*Funding.* Article processing charge is provided by the University of Graz.

*Acknowledgments.* We acknowledge the financial support for article publication by the University of Graz. We sincerely thank all providers of data. The UAV data was provided by the University of Graz. ERA5 data were supported by European Centre for Medium Weather Forecasting (ECMWF). Observational data from the Greenland Ecosystem Monitoring Programme were provided by Department of Bioscience, Aarhus University (Denmark), Department of Geosciences and 830 Natural Resource Management, Copenhagen University (Denmark), Asiaq–Greenland Survey (Greenland), Geological Survey of Denmark and Greenland (GEUS) (Denmark), and Greenland Institute of Natural Resources (Greenland). We are grateful to Brice Noël for his help in providing the Regional Atmospheric Climate Model (RACMO) dataset and related detailed information. We would also like to express our deep gratitude to the Editor and two anonymous reviewers for their valuable time and for their constructive comments and suggestions.

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
