# Peer review of "The importance of regional sea-ice variability for the coastal climate and near-surface temperature gradients in Northeast Greenland"

_EGUsphere, 2023_

## Author Comment (AC1)

**Response to reviewer 1**

**"Coastal climate variability in Northeast Greenland and the role of changing sea ice and fjord ice"**

Shahi et al

Dear Reviewer,
We are very grateful for your very constructive reviews and appreciate the valuable time put into this. By incorporating the reviewer's suggestions, we are confident of having achieved a much more mature manuscript which we hereby submit for your consideration.
In the following, we mark red the comments given by the reviewer, give our answers and comments in black, and indicate how we addressed the amendments in the manuscript in green.
– on behalf of the author team,
Sonika Shahi

Summary

"Coastal climate variability in Northeast Greenland and the role of changing sea ice and fjord ice" by Shahi et al presents a process-level analysis of the local and large-scale drivers of slope temperature gradients (STG) measured at weather stations around the Zackenberg station array. The authors additionally use passive microwave sea ice observations, atmospheric data from UAV retrievals, and ERA5 reanalysis and the Regional Atmospheric Climate Model (RACMO) data fields to evaluate complex associations between nearby Greenland Sea and fjord ice cover conditions, upper-air circulation patterns, and STGs and their impacts on surface mass balance of the adjacent A.P. Olsen (APO) ice cap. Toward understanding these interactions, key findings include high offshore fjord and Greenland sea ice fraction (SIF) is linked with near-surface inversion-like conditions (shallow STG), less snowfall/ablation at APO under anticyclonic upper-air conditions aloft. In the case of low SIF, the opposite associations are found with steep STGs, more snowfall and ablation under a lower pressure/cyclonic regime in the mid-troposphere.

Changes to the regional cryosphere do not typically occur in isolation (i.e., such as local glaciers melt during a warm spell while nearby sea ice does not and vice versa), and the authors put forth a commendable effort to link changes in glacier surface mass balance (SMB) and associated surface-atmosphere interactions within an observation-rich region at Zackenberg. That said, I offer a few suggestions to draw attention more clearly to the paper's themes. In particular, the authors might re-consider if the title reflects the narrative. The authors may also consider what role a changing sea ice/fjord ice cover through time plays in their results. More specific recommendations and minor comments are listed below by line (L) number referencing the submitted version of the manuscript.

We would like to thank the reviewer for his/her time and the positive comments. We have considered all the comments and have modified the manuscript in response. We also adapt the title accordingly as: "The importance of regional sea-ice variability for the coastal climate and near-surface temperature gradients in Northeast Greenland".

General Comment on the Analysis

1) Much like other Arctic marginal seas, summers of the last decade have seen a decline in Greenland Sea ice conditions, which contrast earlier analysis years of the late 1990s and early 2000s when sea ice was more prevalent, for example. Does this change in the ice-coverage and its seasonality (timing of melt and freeze onset) across time impact the interpretation of high versus low sea ice fraction years with regards to the atmospheric processes examined?

That is an interesting and important question and comes up in a similar form yet different wording from Reviewer 2. We put more effort into this topic while initiating this study and plotted the daily sea ice time series (SFig4) to see if the high and low sea ice days are randomly distributed over the study period. We concluded that the distribution was not skewed and rather random, so we did not emphasize it in the initial submission. However, since there is a trend visible in DJF and MAM, we used three different methods to identify breaks when the statistical properties of sea ice potentially changed:

1. We manually selected the middle year of the total period (1996–2020) as a change point i.e., the year 2009, and calculated the atmospheric composites before and after 2009 with respect to sea ice variability.
2. We detrended the sea ice time series using the estimate of the trend component; trends were estimated by applying the Mann-Kendall test at the 5 % significance level and trend strengths were quantified by applying Sen's slope estimator. Then, we calculated the atmospheric composites based on the high (more than 95th percentile) and low (less than 5th percentile) detrended sea ice values (Fig. R1).

[Figure]

**Figure R1. The time series of daily fractional sea-ice cover (SIF in %) zonally (over all grid cells) estimating central tendency (median) over the Greenland Sea for the given season (1996–2020). The upper row**

represents the time series of daily SIF, the middle row represents the detrended daily SIF ($\widehat{SIF}$) time series, and the bottom row represents the daily SIF time series with high and low SIF days determined from the $\widehat{SIF}$. The dots represent the SIF and $\widehat{SIF}$ for all day (black dots), high SIF/SIF days (red dots; days when SIF and $\widehat{SIF}$ are more than their respective 95th percentile), and low SIF/SIF days (blue dots; the days when SIF and $\widehat{SIF}$ are less than or equal to their respective 5th percentile) over the period 1996–2020. The red and blue dashed lines represent the 95th and 5th percentiles of SIF and $\widehat{SIF}$ for the given season. The green line represents the trend line and the estimate of the trend (slope and p value) are represented in green text.

3. We applied an offline change point detection algorithm, Kernel change point (Celisse et al., 2018) to detect a significant change in the statistical properties of fractional sea-ice cover (Fig. S5).

We got a similar pattern of SIF anomalies from all three methods, which supports our results and conclusions in the first version of the paper. As the third method is objective and thus more robust, we included only that method in the manuscript (see Section 2.9.2 in the manuscript for detail).

Irrespective of the subperiod identified and used, we find similar patterns of atmospheric composites based on high and low SIF days. Though the mean value of atmospheric variables changed over time due to climate change, the anomalies remain consistent. Thus, the interpretation of the atmospheric process does not change with high and low SIF.

We argue that with climate change and the related evolution of sea ice, the mean atmospheric condition will change as well, however, the anomalies pattern with respect to the newly defined mean (of the subperiod) will not change significantly. So, in the future, due to climate change, decreased sea ice conditions will be more prevalent, which we add to the result (see Section 3.3.3) and discussion (L753–766) of the revised manuscript.

Specific Comments within Sections

L11-25: In the abstract, a concluding sentence on implications of this work in a changing climate is needed, especially with regards to nearby Greenland/fjord SIF and local APO SMB changes. To this end, within this section the authors might consider more clearly emphasizing regional cryosphere coupling in line with the key findings of the paper.

We add the following sentence in the abstract:

"…we speculate that with climate change the Zackenberg region's atmospheric conditions like a decrease in atmospheric stability and surface mass balance (especially in summer), and an increase in precipitation governed by declining sea ice will be more prominent in the future."

L163-164: Was there a threshold established (>+/- x sigma) for establishing these unrealistic data points? Please clarify.

Visual inspection was carried out by investigating the sub-daily time series plot for obvious contextual outliers. We believe using a fixed in the non-stationary time series can lead to mistakenly identifying data as an outlier.

L185: Please be more specific about this "information" - what local camera satellite products and space-time resolutions are used?

With "information" we meant the dates of the fjord ice formation and break up. We modified the text and added detailed information about the camera time resolution as (L185–188):

"The dates for the onset and breakup of fjord-ice in outer Young Sound are mainly based on an automatic digital camera system (74.3° N, 20.2° W), (Rysgaard et al., 2009). The camera is programmed to take one photo a day at 13:20 all year round. In the years when the camera malfunctioned (only a few times) the data is validated with satellite observations."

However, we did not get further information about the satellite products used upon the correspondence with the programme manager. Since fjord ice conditions change very rapidly within a season from entirely ice-covered to entirely ice-free, and since we take daily resolution as the basis, we assume no dependence on image resolution from ground-based camera as even a very poorly resolved camera will give very similar binary results.

L297-299: Was the composites sensitivity to these STG thresholds tested? It would be a good idea to comment on how the results change, if at all, by using different STG strength criteria.

As mentioned in Lines 314–315, in addition to the mean ± half standard deviation as a threshold definition for high vs. low STGs, we used the 25th and 75th percentiles to compute the composite anomaly. With both criteria, the composite anomaly patterns were very similar (in terms of sign: positive or negative (Table R1 and Table S4). This is simply due to the distribution of the data. All the days collected by using the 25th and 75th percentiles have large absolute magnitudes compared to the days collected using the mean ± half standard deviation, hence, resulting in more negative/positive anomalies. For example, the composite anomaly of wind speed for M3 and ZAC stations shows a similar pattern irrespective of the criteria used (Table R1). In order to keep the focus of the manuscript we do not present these results in detail.

**Table R1: Anomalies of near-surface variables like wind speed ($U_{2m}$ in m s$^{-1}$) measured from AWS corresponding to high and low slope temperature gradient between M3 and ZAC (STG$_{M3-ZAC}$ in °C km$^{-1}$) days for the given season.**

| | | DJF | | MAM | | JJA | | SON | |
|---|---|---|---|---|---|---|---|---|---|
| | | High | Low | High | Low | High | Low | High | Low |
| M3 | STGμ ± 0.5STGσ | -1.6±0.1 | 2.7±0.4 | -0.9±0.1 | 1.3±0.4 | -0.5±0.1 | 0.7±0.2 | -1.3±0.1 | 1.5±0.3 |
| | 25th and 75th STG | -1.6±0.1 | 3.8±0.5 | -0.9±0.1 | 1.9±0.5 | -0.5±0.1 | 0.9±0.3 | -1.3±0.1 | 2.0±0.4 |
| ZAC | STGμ ± 0.5STGσ | -0.9±0.1 | 1.5±0.3 | -0.5±0.1 | 0.8±0.2 | -0.5±0.0 | 0.6±0.1 | -0.7±0.1 | 0.8±0.2 |
| | 25th and 75th STG | -0.9±0.1 | 2.2±0.3 | -0.5±0.1 | 1.2±0.2 | -0.5±0.0 | 0.8±0.2 | -0.6±0.1 | 1.0±0.2 |

As an independent test, we also used a second station pair e.g., M6 and ZAC, which results in the same conclusion that more negative STG$_{M6-ZAC}$ are associated with higher (lower) wind speed at both stations (Table R2).

**Table R2: Anomalies of near-surface variables like wind speed ($U_{2m}$ in m s$^{-1}$) measured from AWS corresponding to high and low slope temperature gradient between M3 and ZAC (STG$_{M3\text{-}ZAC}$ in ºC km$^{-1}$) days for the given season.**

| | | DJF | | MAM | | JJA | | SON | |
|---|---|---|---|---|---|---|---|---|---|
| | | High | Low | High | Low | High | Low | High | Low |
| M6 | STGμ ± 0.5STGσ | -1.1±0.9 | 2.7±1.2 | -0.6±1.2 | 2.6±1.5 | -0.3±0.5 | 1.5±0.9 | -0.9±0.9 | 3.2±1.5 |
| ZAC | STGμ ± 0.5STGσ | -1.1±0.1 | 2.0±0.6 | -0.3±0.1 | 1.0±0.4 | -0.4±0.1 | 0.9±0.4 | -0.6±0.1 | 1.3±0.5 |

We considered the subjectiveness of the choice of thresholds by both applying different thresholds and different station pairs. While the absolute values naturally change slightly, the sign and the quantity and thus the conclusions remain the same, which is why we opted for a simple threshold scheme.

L321: Are the SIF and STG also non-normally distributed with respect to time also? In other words, is there a year within the respective (seasonal) time series where the mean and/or variance change, and how do composites before/after these potential changepoint(s) affect interpretation of results presented here? Greenland Sea ice cover has changed in terms of freeze/melt onset (see Stroeve et al., 2017 cited in manuscript) and extent through much of the annual cycle (see Peng and Meier, 2018, Annals of Glaciology, doi:10.17/aog.2017.32), so some indication on how the SIF evolution with time affects the frequency of low/high SIF with respect to STG, SMB and related processes should be discussed.

We think this question closely relates to the general comments.

That is a very interesting question and we investigated it further in more detail.

STG is normally distributed; however, SIF is not normally distributed. Keeping the distribution of the data in mind (as mentioned in the manuscript) we also used the 25th and 75th percentile of the STG to test its sensitivity. Since there are data gaps in the station data, it is not recommended to perform the change point detection method in STG. If done so, there is a high chance of getting false positive change points, which is why we only applied the change point detection method in the SIF data.

L454-461: The u10m direction could also be emphasized in this section (I could be wrong, but my understanding is +u = westerly & offshore (in this case), -u = easterly & onshore).

Thank you for the suggestions. The impact of the zonal(u) and meridional(v) wind components on STG is an interesting topic and we briefly explored these wind components before; however, we did not include this part in the manuscript in detail, but we refer to Figure S3 for general prevailing wind patterns.

L477/Figure 8: Might it make sense to flip the color ramp such that more ice (colder conditions) are blue hues and less ice (warmer conditions) are red hues? Figure 11 involving APO SMB applies this rationale to its anomaly SMB color ramp.

Thank you so much for the suggestions. We modified the figure as suggested.

Technical Corrections

L15: near-fjord sea or land ice conditions?

We mean surface type as the presence or absence of snow and near-fjord ice as the presence or absence of ice in Young Sound only. So, we keep the text as it is.

L19: Would suggest removing "Evidently" and start sentence with "A positive…"

Thank you for the suggestions. Modified as suggested.

L21: Instead of "change" I'd recommend adding a descriptor (i.e., reduction)

Modified as suggested.

L67: In the atmospheric science community, zonal references east-west and meridional north-south, so would consider switching terms here for the sake of clarity.

Thank you for the suggestions. Modified as suggested.

L114-116: I'd suggest combining the two sentences, such as "…the present study examines the statistical relationships between STG, SIF, and other atmospheric variables and their physical connections."

Thank you for the suggestions. Modified as suggested.

L119: The year range of ZR temperature monitoring could be listed here.

Thank you for the suggestions. Modified as suggested.

L126: Established by whom (ClimateBasis?) and when in 1995? The following sentences mention 1996 hence clarifying the starting year and month.

Thank you for the suggestions. Modified as suggested.

L136-137: Suggest change "for the entire" to "across Greenland"

Modified as suggested.

L152: Suggest modification to "These datasets provide…"

Thank you for the suggestions. Modified as suggested.

L207: Instead of cross do you mean "x" marker?

Yes. Modified as suggested.

L281-283: This is a bit ambiguous; are there numerical values/ranges associated with steep and shallow STGs?

We use steep and shallow STGs as a comparative vocabulary. We added one example in the manuscript (L285–289) to clarify the use of these terminologies:

"The terminology was used as follows: for a given STG, steeper STG < STG < shallower STG. For example, if the ELR is -6.5 °C km$^{-1}$ and a STG between AP2 and AP1 stations (STG$_{AP2-AP1}$) is -7 °C km$^{-1}$, and between M3 and ZAC stations (STG$_{M3-ZAC}$) is -5 °C km$^{-1}$ then ELR is shallower than STG$_{AP2-AP1}$ whereas steeper than STG$_{M3-ZAC}$. Additionally, we refer to more and less positive STG as 'strong' and 'weak' inversions, respectively."

L347: "During the UAV measurement period?..."

Modified as suggested.

L353-354: If possible, would recommend an estimate of this "same elevation" value.

The information about the elevation is added. It is the elevation of the AWSs but above the ground level: for ZAC 43 m a.g.l. and for M3 420 m a.g.l..

L357: Do you mean "screen-level" on the temperature instrument? Please clarify.

By screen-level temperature, we mean 2 m air temperature. We added this information to the manuscript (L384–385):

"This highlights the potential of AWSs to capture the inversions in the ZR, despite the surface characteristics influencing screen-level (2 m) temperature."

L378: Does "early" need to be in parenthesis here?

We removed the parenthesis.

L380: Do you mean total spring days? Please clarify here and similarly in the sentence that follows.

Yes, a total of 1131 spring days were available for both M3 and ZAC stations. Similarly, 1440 summer days overlapped (and were available) for both stations.

Modified as suggested.

L395/Figure 5: Might a colorbar be added to the figure to depict the SIC gradient (faint to dark blue) that evolves during the freeze-melt periods?

Thanks a lot for the suggestions. We agree that stating "transitional period" has been misleading. Instead of the shading we used in the initial submission that in principle gave an indication of relative sea ice coverage for a given day based on observations during the study period without

showing numbers, we add quantitative information in the revised version. We calculate the fraction of occurrence of fjord ice on a given day. For example, a value of 80 % on a day of the year 172 means that 80% of all 21st June during the observation period had fjord ice cover.

L432: Remove comma after "snow" and suggest removing "and as expected," also

Modified as suggested.

L443-444: Would suggest removing this sentence as it does not add substantive summary of the previous results described.

Thank you for the suggestion. However, we consider not removing this sentence as it states the general linkage between synoptic scale condition (from RACMO) and local scale measurements (from AWS).

L450: GrSea sea ice edge averaged for the full period? Please clarify what the dashed black line shows in each panel.

The dashed black line represents the Greenland Sea as defined by the International Hydrographic Organization (available online at http://www.marineregions.org/.). This information is provided in Figure 1. However, for the sake of readability, we added this information to all figures.

L512 and 562-563: "~27 change in SIF" – is this % change? There are a few instances in the paper where sea ice fraction (SIF) is not associated with units.

Thank you so much for pointing out this mistake. We added % after the number.

L564: "…greater impact on the air close to the surface than higher above" such as at what atmospheric layer(s) according to your analyses?

To study the change in the vertical temperature gradient in response to sea ice variability, we use air temperature at five (available) pressure levels (1000 hPa, 925 hPa, 850 hPa, 700 hPa, and 500 hPa) from RACMO (as defined in the Data and method section of the manuscript and in Figure 10). The impact is greater at 1000 hPa (close to the surface) than at the layers above (up to 500 hPa). We added this information to the manuscript (L591–592):

"This result suggests that the variability of sea ice has a greater impact on the air close to the surface (at 1000 hPa) than higher above (at 500 hPa)."

L675: Synoptic-scale time lag of the processes? It is good to acknowledge this timescale as it is relevant to the Stroeve et al. (2017) process interpretation and likely the same timescale of processes that interact in your analysis.

Thank you for the suggestions.

This is a local-scale influence time lag. Stroeve et al. (2017) found a correlation between the timing of melt onset, which occurs on average nine days earlier over the sea ice than on the adjacent ice sheet.

We modified the manuscript accordingly (L740–744):

"However, unlike Stroeve et al. (2017), we did not account for the time lag (1-week lag) effect of the heat transfer from reduced sea-ice areas further inland to the GrIS. This might influence the magnitude of the SMB anomaly. However, considering the short distance between APO and water bodies i.e., Young Sound (~35 km) and the outer coast (~75 km), we speculate a shorter time lag between sea ice melt and ice cap response (though elevation might intervene the interaction)."

L679: Change to "According to the authors,…"

Modified as suggested.

---

## Author Comment (AC2)

**Response to reviewer 2**

**"Coastal climate variability in Northeast Greenland and the role of changing sea ice and fjord ice"**

Shahi et al

Dear Reviewer,
We are very grateful for your very constructive reviews and appreciate the valuable time put into this. By incorporating the reviewer's suggestions, we are confident of having achieved a much more mature manuscript which we hereby submit for your consideration.
In the following, we mark red the comments given by the reviewer, give our answers and comments in black, and indicate how we addressed the amendments in the manuscript in green.
– on behalf of the author team,
Sonika Shahi

**General comments**

Shahi and coauthors analyze the factors controlling the vertical temperature structure of the atmosphere over the Zackenberg region of northeast Greenland. They utilize a dense network of weather stations arrayed throughout varying elevations in this topographically complex area, along with UAV observations, atmospheric reanalysis, and regional climate model output. Their main findings are: (1) atmospheric inversions - characterized by a more "shallow" slope temperature gradient between the low- and high-elevation stations - are more common during the cold season and less frequent during the summer; (2) inversions are associated with anomalously high 500 hPa geopotential heights and regional sea ice cover; and (3) the influence of these environmental conditions on the surface mass balance of the A. P. Olsen Ice Cap are complex, particularly in the summer when lower sea ice fraction is associated with more melt at lower elevations but more mass gain through accumulation at higher elevations.

This is a interesting study overall and the authors have done a nice job of integrating their rich observational dataset with other contextual data sources to understand the environmental controls on atmospheric temperature profiles in the region. The UAV observations are novel in this part of Greenland, to my knowledge, and are a good complement to the station data. The references are thorough and relevant. However, I do have comments on several aspects of the paper that should be improved to make it suitable for final publication. The writing style is often difficult to read with extensive use of acronyms and parenthetical asides, and I think the results could be communicated more effectively by using more plain-language descriptions where possible. I am also not completely convinced that the authors' assertions about the causative influence of sea ice conditions on the regional atmospheric conditions are robustly supported by their analyses. More detail on these comments and some additional comments are provided in the specific comments and technical corrections.

We would like to thank the reviewer for his/her time and the positive comments. We agree with the comments made and have modified the manuscript in response. We remove the abbreviation for the word which does not repeat more than three times. We also provide a detailed comment on the causative influence of sea ice and that we formulate it more as a hypothesis in the revised

manuscript while indicating how this could be approached in future studies, mainly through high-resolution modeling and sensitivity studies associated.

**Specific comments**

In my opinion the writing style in the manuscript should be more clear. The writing is burdened by excessive use of acronyms, parenthetical phrases, and repeating statements of the same fact. These factors disrupt the flow of the paper. One example is the sentence from L16-18 sentence in the abstract: "...we find that shallow, i.e. more positive (inversions) or less negative than the mean condition, STGs are associated with a positive anomaly in geopotential height at 500 hPa and surface pressure over East Greenland" - it seems to me this could be simply written as "we find that temperature inversions are associated with positive 500 hPa geopotential height and surface pressure anomalies".

We followed the advice and reformulated the specific sentence accordingly. Furthermore, we went through the manuscript and disentangled some of the admittedly complicated sentences. We think the manuscript reached a clearer writing style. If needed, a comprehensive list of abbreviations will be added that should also help make it more readable.

Regarding acronyms, I understand it is challenging to strike a balance between excessive acronym use on the one hand and not using the same full phrase repeatedly on the other. However, I think the authors should consider which acronyms are truly necessary and substitute concise plain-language descriptions where possible. For example, the sentence from L629-630 would be more readable if written as something like "Figure 12 shows the comprehensive picture of the potential relationship between the vertical temperature structure, mid-tropospheric circulation, sea ice cover, and ice cap surface mass balance."

Thanks for the concrete suggestion. We follow the advice and as stated above, reduce the complicated writing wherever possible. As aforementioned, we decided to follow the advice by omitting acronyms that occur fewer than three times.

As a related comment, I found the language used to describe slope temperature gradients confusing, especially when trying to think about the slope temperature gradient in the context of inversions. For example, in L455-456 and at many other points, the authors use the terms "steeper (shallower)" to refer to the slope temperature gradient. This is confusing because the slope temperature gradient is directly related to atmospheric inversions, but inversions are themselves also often referred to as "shallow" (with a completely different meaning to the word in this context). And at first glance, one would think the term "steeper" would refer to a stronger inversion, but it is actually the opposite, as "steeper" STGs mean there is a greater decrease in temperature from the surface to the higher-elevation stations. The authors also throw in the term "positive STG" (e.g. L460) to refer to the stronger inversion / shallower STG conditions. I suggest substituting plain-language physical descriptions for acronyms in at least some places to remind the reader of the physical meaning of the STG variable - i.e. simply state if there is an inversion present or, alternatively, if the temperature decreases with height.

Thanks a lot for the suggestion. We modified all the text as suggested and add instances where it is suitable for plain-language versions that indeed are close to the physical meaning of STG.

Reviewing literature extensively we found this very problem to be challenging as different authors apply different wording. We thus adhere to the definition as stated in L281–287 of the revised version and added one example to clarify the use of these terminologies and some plain-language wording wherever applicable.

We added one example in the manuscript (L285–289) to clarify the use of these terminologies:

"The terminology was used as follows: for a given STG, steeper STG < STG < shallower STG. For example, if the ELR is -6.5 °C km$^{-1}$ and a STG between AP2 and AP1 stations (STG$_{AP2-AP1}$) is -7 °C km$^{-1}$, and between M3 and ZAC stations (STG$_{M3-ZAC}$) is -5 °C km$^{-1}$ then ELR is shallower than STG$_{AP2-AP1}$ whereas steeper than STG$_{M3-ZAC}$. Additionally, we refer to more and less positive STG as 'strong' and 'weak' inversions, respectively."

I am not sure the authors have convincingly proven that the correlation between regional sea ice and atmospheric conditions in the Zackenberg region represents a direct causative influence of sea ice on the atmosphere. For example, the authors find a seasonal relationship between Greenland Sea sea ice fraction and Zackenberg temperature and precipitation (L644-646). I agree that it is very likely that reduced sea ice contributes to atmospheric warming and moistening. However, could it not also be the case that the sea ice and atmospheric conditions are both influenced by the same large-scale circulation patterns, leading to a correlation between sea ice and atmospheric conditions that is not a one-directional causal influence of sea ice on the atmosphere?

We acknowledge this comment and appreciate this as it inspired us to rethink some of our hypotheses. Such causative connections indeed can run both ways and in the revised version we add this important point explicitly (in the discussion section L762–766) by stating:

"We acknowledge that parts of the connections shown in Figure 12 may indeed also apply in reverse order or are driven by the same large-scale conditions, e.g., higher air temperatures and low sea ice conditions are intrinsically related. While local-scale processes influence the melt variability, the large-scale atmospheric circulation (500 hPa geopotential height anomalies) can be a superimposed driver in controlling both the sea ice and ice cap melt (Stroeve et al., 2017). However, the clear association between sea ice anomalies and nearby coastal climate anomalies indicates a local response."

In essence, the compositing method contrasts high SIF days with low SIF days and relates them to the corresponding atmospheric conditions. Because there is no reason to expect a linear relationship between variability in sea ice coverage and STG (atmospheric conditions), a compositing technique is preferred over linear correlation. This is one of the advantages of our method and why we used this technique. If we opt for multiple linear regression, we face problems due to multicollinearity within the data set. Furthermore, applying partial correlation as used in Stroeve et al. (2017) to remove the influence of the Greenland Blocking described by GBI, as an indicator of the prevailing large-scale atmospheric circulation over Greenland on both SIF and atmospheric variables would be challenging. This is because GBI merely measures the

average geopotential height at 500 hPa pressure level over Greenland, and not the types or locations of the blocking (Preece et al., 2022), which affects regional climate differently.

In order to assess whether subjectivity is introduced by the choice of SIF anomaly domain, we also computed the composite based on SIF data close to the Zackenberg region (SIF averages over the black rectangle in Figure 1). The results were consistent with the composites derived from the whole Greenland Sea scaled SIF data. The atmospheric composite anomaly derived for this SIF evidently reflects the local scale (direct) influence. Also, given the vicinity of the Zackenberg region to the fjord (~35 km) and the ocean (~75 km), we expect an impact of moisture and heat transfer.

We think that our study builds the motivation for future studies to actually address these topics by applying dynamical modelling experiments using different boundary conditions and drivers.

Similarly, do the authors account for the climatological evolution of sea ice conditions within seasons in their analyses? For example, the authors find that, on a daily time scale during the summer, reduced sea ice is correlated with reduced SMB in the ablation area. Again, I agree that the reduced sea ice conditions likely influence atmospheric warming, but could it not also be the case that both sea ice cover and SMB are correlated with the climatological pattern of warming temperatures throughout the course of the summer? I apologize if the authors have already addressed these questions in their methodology, however I could not find any reference to these issues in Section 2.9.

Regarding the implication of the climatological evolution of sea ice, that is a very interesting question and comes up in a similar form yet different wording from Reviewer 1. We put more effort into this topic while initiating this study and plotted the daily sea ice time series (SFig4) to see if the high and low sea ice days are randomly distributed over the study period. We concluded that the distribution was not skewed and rather random, so we did not emphasize it in the initial submission. However, since there is a trend visible in DJF and MAM, we used three different methods to identify breaks when the statistical properties of sea ice potentially changed:

1. We manually selected the middle year of the total period (1996–2020) as a change point i.e., the year 2009, and calculated the atmospheric composites before and after 2009 with respect to sea ice variability.
2. We detrended the sea ice time series using the estimate of the trend component; trends were estimated by applying the Mann-Kendall test at the 5 % significance level and trend strengths were quantified by applying Sen's slope estimator. Then, we calculated the atmospheric composites based on the high (more than 95th percentile) and low (less than 5th percentile) detrended sea ice values (Fig. R1).

[Figure]

**Figure R1.** The time series of daily fractional sea-ice cover (SIF in %) zonally (over all grid cells) estimating central tendency (median) over the Greenland Sea for the given season (1996–2020). The upper row represents the time series of daily SIF, the middle row represents the detrended daily SIF ($\widehat{SIF}$) time series, and the bottom row represents the daily SIF time series with high and low SIF days determined from the $\widehat{SIF}$. The dots represent the SIF and $\widehat{SIF}$ for all day (black dots), high SIF/SIF days (red dots; days when SIF and $\widehat{SIF}$ are more than their respective 95th percentile), and low SIF/SIF days (blue dots; the days when SIF and $\widehat{SIF}$ are less than or equal to their respective 5th percentile) over the period 1996–2020. The red and blue dashed lines represent the 95th and 5th percentiles of SIF and $\widehat{SIF}$ for the given season. The green line represents the trend line and the estimate of the trend (slope and p value) are represented in green text.

3. We applied an offline change point detection algorithm, Kernel change point (Celisse et al., 2018) to detect a significant change in the statistical properties of fractional sea-ice cover (Fig. S5).

We got a similar pattern of SIF anomalies from all three methods, which supports our results and conclusions in the first version of the paper. As the third method is objective and thus more robust, we included only that method in the manuscript (see Section 2.9.2 in the manuscript for detail).

Irrespective of the subperiod identified and used, we find similar patterns of atmospheric composites based on high and low SIF days. Though the mean value of atmospheric variables changed over time due to climate change, the anomalies remain consistent. Thus, the interpretation of the atmospheric process does not change with high and low SIF.

We argue that with climate change and the related evolution of sea ice, the mean atmospheric condition will change as well, however, the anomalies pattern with respect to the newly defined mean (of the subperiod) will not change significantly. So, in the future, due to climate change,

decreased sea ice conditions will be more prevalent, which we add to the result (see Section 3.3.3) and discussion (L753–766) of the manuscript.

Regarding the second question, we believed that this comment is replied to in the extensive comment above.

The Introduction does a nice job of explaining why the vertical structure of temperature (and not just temperature in general) is important to study. I think some of this language should be adopted in the Abstract to explain the overall objectives and importance of the study, instead of simply stating that the purpose of the study is "to capture this complexity" (L11-12).

This is a good suggestion which we follow by expanding the sentence in the Abstract to:

"Due to its sensitivity to ecosystem components, it is crucial to capture this complexity by quantifying seasonal variability of stability conditions and regional cryosphere coupling. This study use..."

As a related comment, I don't think the title accurately captures the contents of the study. The study primarily concerns the controls of large-scale circulation and regional sea ice cover on the *vertical temperature structure* of the atmosphere in the Zackenberg region, but the vertical temperature profile aspect is not contained in the title.

We agree and adapt the title accordingly.

"The importance of regional sea-ice variability for the coastal climate and near-surface temperature gradients in Northeast Greenland"

Fig. 1 is a nice overview map and helps the reader understand the spatial setting of the study.

Thank you for the positive feedback.

L84: Should "equatorward" be "poleward" here? I.e. the warm air mass is located in an anomalously poleward area due to the blocking?

We meant the warm air mass is trapped equatorward of an 'anticyclonic ridge' though it is near to the poleward area due to the blocking. However, we changed the sentence for clarity to (L87–88):

"During the blocking event, the polar jet streams develop nearly stationary meanders and trap warm air in an anomalously poleward area (Sirpa et al., 2011; Hanna et al., 2015)."

L96-97 and elsewhere: Is it necessary to use the "SIF" abbreviation so frequently? At least in some places the text could just name sea ice as "sea ice" rather than SIF.

As per the suggestion we replaced SIF with sea ice wherever applicable. We used the term "sea ice" in general comments when there is no need to be specific about the sea-ice characteristics. And we used SIF to be clear about which characteristics of sea ice we are referring to.

L99: Is the abbreviation "GrSea" really necessary?

We removed the abbreviation of the Greenland Sea and used the full name.

L113-114: I think the A. P. Olsen Ice Cap SMB analysis should be considered objective 3. It is a substantial part of the analysis in the paper. Correspondingly, there should be a separate "section 3.4" starting with L371, instead of grouping the SMB analysis into section 3.3.

Thank you for the suggestion. Indeed the A. P. Olsen Ice Cap SMB analysis is an important part of the manuscript, so we include SMB analysis as the third objective of the paper. Rather than making a separate section 3.4, we divided section 3.3 (Impact of sea-ice variability) into three parts: (3.3.1) Near-surface and atmospheric processes, (3.3.2) Surface mass balance of a local ice cap, and (3.3.3) On the temporal robustness of the relationship between sea-ice evolution and near-surface atmospheric conditions.

L121-122: The geographical description of "west" vs. "east" is confusing here. So the ZR is ~40km west of the outer coast and ~70km east of the GrIS?

Yes. We revised the sentence for clarity. (L124–126)

"The ZR is located in Northeast Greenland, midway between the outer coast (~40 km to the east of ZR) and the GrIS (~70 km to the west of ZR)."

L134: It would be helpful to go ahead and state the elevation of ZAC and M3 stations in this sentence. As is, the reader must reference Table 1 and Figure 2 on the subsequent pages to figure this out.

Thank you for the suggestions. We added the stations' elevation information in the text (L138–140).

"For the period 2004–2018, in between ZAC (43 m a.s.l.) and M3 (420 m a.s.l.), (where M3 is representative of the area above the inversion layer), inversions are frequent (70–75 %) during winter, whereas during summer the inversion frequency decreases (27–58 %)."

L133-137: Is there spatial variability in the characteristics of inversions in the ZR? Or is the assumption that the stations are sufficiently near to each other that they are vertically sampling the same air mass?

This is a good question we are not entirely able to answer. However, given the vicinity of the stations in the valley (most of the stations are installed within the 2–3 km wide valley), we assume they are sampling the same air mass with minor spatial variability. We are confident that the horizontal spatial variability is much less than the vertical gradients.

Figure 3: The gray shadings of the UAV vertical profiles in the JJA panel for 2017 and 2018 are difficult to distinguish from one another. Perhaps a color other than gray could be used for one of the years?

We consider the importance to distinguish the mean vertical temperature profile for different periods which is clearly discernable from the chosen color; black for 2017 and grey for 2018. The shadings which represent the 25th and 75th percentiles of the given temperature profiles are around the respective profiles, completely separated from each other. So, we consider keeping the figure as it is to avoid overuse of the colors.

L218: Psurf is not an upper air variable.

Thank you for pointing it out. We removed $P_{surf}$ from the sentence.

L226-230: Have the authors considered using sea ice data from a higher-resolution source when available, for example AMSR data (Meier et al. 2018) from 2012-present?

Using higher-resolution sea ice data sounds interesting. However, since sea ice data from ERA5 is prescribed in RACMO and we used RACMO surface and atmospheric products to study the synoptic scale processes, we consider not using an independent dataset that needs further evaluation. Also, we consider our results are not dependent on the spatial resolution of the sea ice data. So even after using high-resolution sea ice data, we believe it will not compromise our main finding.

L252-262: Nice idea to evaluate RACMO over ice-free surfaces and compare to ice-covered, this is a novel contribution of the study.

Thank you for your positive feedback. Indeed, we believe this is novel and relevant.

L343-357: UAV profiles

- Where were the UAV profiles taken in relation to stations M3 and ZAC? Can the locations be marked on the map in Figure 1?

UAV profiles were taken within 10 m from the ZAC station. We mentioned it in the manuscript in L195–198. To keep the readability of Figure 1, we just added the UAV flight information in the Figure 1 caption and referred to it in Section 2.5.

- What were the temporal sampling characteristics of the UAV profiles? Were the flights in each of 2017 and 2018 performed on the same day? Were the 2017 and 2018 flights performed in similar times of the year (more specific than summer)?

The detailed timing of the UAV measurements is shown in SFig. 1(b). As shown in the figure, the temporal sampling of the UAV profile is not consistent, and the flights were not performed on the same time and day of the year as they developed in a campaign and hence contain an opportunistic sampling approach when it comes to the period of the year. The time of the day

was consistent throughout the campaigns and hence increases comparability. In 2017, the measurement was carried out from 30th May to 6th June; in 2018, from 2nd–3rd August and 19th–21st August.

Figure 5: The plot lines and confidence interval shading are difficult to distinguish from the sea ice background shading. Consider different color choices or perhaps plotting sea ice as a line rather than shading. If keeping the shading, the color shading should probably be mapped to specific values rather than a non-quantitative "transitional period" - there is no way for the reader to discern what sea ice values this phrase refers to.

Thanks a lot for the suggestions. We agree that stating "transitional period" has been misleading. Instead of the shading we used in the initial submission that in principle gave an indication of relative sea ice coverage for a given day based on observations during the study period without showing numbers, we add quantitative information in the revised version. We calculate the fraction of occurrence of fjord ice on a given day. For example, a value of 80% on a day of the year 172 means that 80% of all 21$^{st}$ June during the observation period had fjord ice cover.

Figures 6-7: These maps are visually busy and difficult to read. I suggest using a color other than white for the coastlines so they can be more easily seen. Are the dots and the black mesh both necessary?

Thanks a lot for the suggestion. We tried several colors to make the plot easy to read, however, we found the white color as the best option given the colorful background. To improve readability, we increased the thickness of the coastline for better visibility.

The white dots and black mesh represent 0.05 and 0.1 significance levels respectively. We use these two contrasting colors to make sure the significant area is visible irrespective of the light or dark color in the background. So, we consider keeping both. To address the advice, we reduced the density and width of both hatches for better visibility.

Figures 6-11: What is the climatological reference period from which the anomalies are calculated? Is it 2004-2019 as mentioned in the Figure 6 caption, or 1996-2020 as mentioned in section 2.9.2?

The baseline reference period for Figures 6, 7, and 8 is 2004–2019. These plots represent the atmospheric composites based on the slope temperature gradient between M3 and ZAC station. For this station pair, data is available for the period 2004–2019.

The baseline reference period for Figures 9, 10, and 11 is 1996–2020. We start from the period 1996 to match the beginning of the monitoring program in the Zackenberg region. These plots represent the atmospheric composites based on the sea ice in the Greenland Sea.

We refer to this in the respective figure captions.

L522: It would be nice to provide a little more detail in what mesoscale forcing drives the northerly wind during the winter, like the explanation the authors provide for the onshore winds during summer.

Thanks for the suggestion. We added this information in the manuscript (L549–551) as:

"…in winter the north-south pressure difference favors the mesoscale wind field (northerly wind) and will be part of the shallow katabatic outflow from the ice sheet."

L527-534: The result that areas near the coast are warmer than inland during high SIF days is an interesting and unexpected finding.

Thank you for your feedback. It is an interesting finding and can open the door for more detailed research in the future.

Figure 12: I am struggling to wrap my head around the result that higher 500 hPa heights are associated with a *decrease* in ablation, and vice versa. This is counter intuitive because one would expect higher temperatures and greater melt with a 500 hPa ridge overhead. Is the explanation that the Z500 increases are local rather than large-scale in extent (e.g. L630-632)?

Indeed, an increase in temperature leads to an increase in 500 hPa geopotential height (Z500) and could indicate more melt. The relation does not need to be straightforward as it will always be connected to the general conditions and the zero-degree boundary, meaning that even if we find a positive anomaly but the temperatures are very low, this will not necessarily lead to more melt. In that respect, we expect a highly non-linear relation in place. The positive anomaly of Z500 that we find is based on the STG between stations in the Zackenberg region, implying local conditions. During the high $STG_{AP3-AP1}$ when we found a similar Z500 anomaly pattern, for all seasons; we found the effect to be stronger at the higher elevation stations e.g., $T_{2m}$ anomaly is more positive in AP3 (accumulation area, 1475 m a.s.l.), compared to the lower elevation stations e.g., in the AP1 (ablation area, 660 m a.s.l.), (see Figure R2 and Table R1). In Figure 12, the decrease in ablation is indicated in the ablation area where the temperature is showing mostly negative anomaly, hence less melt.

**Table R1: The slope temperature gradient ($STG_s$ in $^\circ$C km$^{-1}$) between AP3 ($T_{2m}$, AP3) and AP1 ($T_{2m}$, AP1) pair and mean anomalies of 2 m air temperature ($T_{2m}$ in $^\circ$C) corresponding to high and low $STG_{AP3-AP1}$.**

| $T_{2m}$ | DJF | | MAM | | JJA | | SON | |
|---|---|---|---|---|---|---|---|---|
| | High | Low | High | Low | High | Low | High | Low |
| $T_{2m, AP3}$ | 0.7±0.9 | -1.2±0.8 | 1.5±1.0 | -1.1±0.8 | 2.3±0.4 | -2.3±0.3 | 1.0±0.8 | -1.3±0.7 |
| $T_{2m, AP1}$ | -3.3±0.9 | 1.8±0.8 | -2.1±0.9 | 1.7±0.8 | 0.5±0.4 | -0.8±0.3 | -2.0±0.9 | 1.0±0.6 |
| $STG_{AP3-AP1}$ | -1.9 | -5.8 | -2.4 | -5.9 | -4.8 | -6.6 | -3.4 | -6.3 |

[Figure]

**Figure R2.** Composite of geopotential height at 500 hPa (Z500 in m) anomalies (shading) ansd means (grey contours; 60 m interval) from ERA5 corresponding to the STG between AP3 and AP1 (STG$_{AP3-AP1}$) for entire daily measurements (number of days indicated at the top of each seasonal column) for a given season.

L671-677: This paragraph about the influence of SIF on SMB is specific to the summer, correct? This should be mentioned here if so.

Yes, it is specific for summer. Modified as suggested.

L686-689: This study also used UAV data.

UAV data was added to the sentence.

In the discussion and/or conclusion, it would be nice to see some discussion about implications of this study for climate change. How will projected future decreases in sea ice extent and potential changes in large-scale atmospheric circulation patterns affect the vertical thermal structure of the atmosphere (i.e. the STG) in the Zackenberg region? What does this mean for the future SMB evolution of Greenland's peripheral ice caps and glaciers (as well as the main ice sheet)?

Thanks for this advice, we added a few lines about the implication of this study for climate change in the discussion section (L753–761) as:

"One of the manifestations of climate change is a decline in the sea ice (Stroeve and Notz, 2018; Peng and Meier, 2018; Polyakov et al., 2022). Our study shows that local change in the sea ice due to changing climate impacts the ZR atmospheric condition. However, even with the observed decline in Arctic sea ice, the consequence for the atmospheric anomaly with respect to high and low SIF remains consistent. Probably with climate change the atmospheric conditions governed by less sea ice (low SIF days) will be more prominent in the future. While we show

that lower sea ice coincides with a decrease in atmospheric stability, we hypothesize a non-linear response on SMB or runoff, which makes predictions out of the scope. In general, decreased stability and increased moisture transport point towards increased precipitation and hence an accelerated water cycle. This is in line with projections of Arctic climate change and should be studied in detail on a regional scale both using observations and model results (McCrystall et al., 2021)."

**Technical corrections**

L81: "anticyclone... which is" --> "anticyclones... which are"

Modified as suggested.

L411 and elsewhere: "moistures" --> "moisture"

Modified as suggested.

L432: "excepted" --> "expected"

Word removed for the sentence.

L512-517 and elsewhere: include units (%) for SIF values.

Modified as suggested.

#

References

Meier, W. N., T. Markus, and J. C. Comiso. (2018). AMSR-E/AMSR2 Unified L3 Daily 12.5 km Brightness Temperatures, Sea Ice Concentration, Motion & Snow Depth Polar Grids, Version 1 [Data Set]. Boulder, Colorado USA. NASA National Snow and Ice Data Center Distributed Active Archive Center. https://doi.org/10.5067/RA1MIJOYPK3P. Date Accessed 03-14-2023.

---

## Author Response (AR2)

**Response to Reviewer 2**

**"The importance of regional sea-ice variability for the coastal climate and near-surface temperature gradients in Northeast Greenland"**

Shahi et al

Dear Reviewer,

We are again grateful for your constructive reviews and appreciate the valuable time put into this. We have revised the manuscript according to your suggestions.

In the following, we mark red the comments given by the reviewer, give our answers and comments in black, and indicate how we addressed the amendments in the manuscript in green.

– on behalf of the author team,

Sonika Shahi

**General comments**

Thanks to the authors for their detailed responses to the reviewer comments. I feel the manuscript is improved in this revision. In particular the physical significance of the STG is more understandable and the manuscript has become easier to read with the edits to the text and removal of some unnecessary abbreviations.

We appreciate your positive comment and the valuable suggestions you provided during the review process which improved the manuscript.

I do have some remaining comments that should be addressed before the manuscript is considered for publication. Most importantly, I am still not following the argument in Fig. 12 that Z500 increases are associated with ablation decreases. It seems to me that this figure is attempting to summarize the authors' findings about the influences of (a) large-scale circulation and (b) local surface type on the ZR climate in a way that is not logically supported by their study framework, as detailed in the specific comments below. I also have a number of requests for clarification and technical corrections, as described in the detailed comments below.

We agree with the comments made and have modified the manuscript in response. We modified Figure 12 accordingly. We provided detailed answers below.

**Specific comments**

I am still not convinced by the authors' argument that Z500 increases are associated with ablation decreases and vice versa, as presented in the schematic diagram in Fig. 12. Their results show that (a) days with a stronger inversion in the ZR tend to have anomalously high Z500, and (b) higher sea ice coverage is associated with a stronger inversion and less ablation. However, it does not necessarily follow from these results that higher 500 hPa heights lead to less ablation, as Fig. 12 implies. It could be the case that the presence of a mid-tropospheric ridge (i.e. higher Z500)

increases temperatures throughout the atmospheric column but with a greater increase at the higher elevation stations - as indeed the authors seem to state in L931-933 - which would manifest as a stronger inversion, but nevertheless a warmer lower atmosphere in absolute terms that promotes enhanced melting during the summer. I suspect that if the authors inverted their composite technique by calculating mean SMB on summer days with above- and below-normal Z500 over the area, they would find that increased Z500 leads to increased melt, contradicting the conceptual framework in Fig. 12.

We agree on the fact that many factors play in, and it is difficult to isolate them into one. For instance, a Z500 anomaly will have different impacts on spatial melt patterns if it occurs below, around, or above the melting point. In response, we modified Figure 12 and removed the feature representing Z500 from the figure and all the references associated with it in the main text to avoid confusion. Furthermore, we simplified other components of the conceptual figure and hope to satisfy the reviewer's criticism. Below we show the updated Figure 12:

[Figure]

Figure 12. A schematic representation of some important linkages in the Zackenberg region (ZR), fractional sea-ice cover (SIF) in the Greenland Sea, and surface condition of A. P. Olsen Ice Cap (APO) especially for the summer season. The temperature at the lower station (LS), $T_{LS}$, can be higher or lower than the temperature at the upper station (US), $T_{US}$. (a) The upper panel shows the conditions when $STG_{US-LS}$ is shallow ($T_{LS}$ is less than $T_{US}$; less negative or positive (inversion) $STG_{US-LS}$), when there is more snow in the valley, when SIF is high, along with low precipitation amounts, and little snow/ice ablation at APO. (b) The lower panel shows the opposite conditions when $STG_{US-LS}$ is steep ($T_{LS}$ is more than $T_{US}$; more negative or less positive $STG_{US-LS}$), when there is less snow in the valley, when SIF is low, along with high precipitation amounts, and strong snow/ice ablation at APO. The solid grey line represents a climatological mean of the temperature profile in the ZR and the corresponding solid black lines represent the mean temperature profile in each condition

The influence of snow cover on STG is a key result described in section 3.1 and Fig. 5. I think it should be mentioned in the abstract and incorporated into the conceptual diagram (Fig. 12).

Thanks a lot for the suggestion. In response, we also added snow cover conceptually in Figure 12. We also mentioned the influence of snow cover on STG in the abstract (L16–17):

"For all seasons, our results show that snow cover and near-fjord ice conditions are the dominating factors governing the temporal evolution of the STG in the Zackenberg region."

Be clear about discussing slope temperature gradient(s) as singular or plural. For example, L14 and the first sentence in L17 are about "slope temperature gradients" (plural), but then the next sentence in L17 switches to "STG" (singular) without an apparent reason for the change, and the rest of the abstract refers to "STG" (singular).

Thanks a lot for the suggestion. In the revised manuscript, we only used the singular form of STG in the abstract.

L11-12: "Due to its sensitivity to ecosystem components" - what does this mean? Does this mean that regional ecosystem processes are sensitive to stability conditions? Please rephrase.

This line paragraph is rephrased as per the suggestion (L11–13):

"Since the regional ecosystem processes are sensitive to atmospheric stability conditions, it is crucial to capture this complexity including adequate cryosphere coupling."

L15-23: Are the results about the local- and large-scale drivers of STG valid for the full year? This should be specifically stated in the abstract, since summer-only results are reported later on in the abstract.

Thanks for the suggestion. We added the time frame for clarity in L16–17:

"For all seasons, our results show that snow cover and near-fjord ice conditions are the dominating factors governing the temporal evolution of the STG in the Zackenberg region."

L20: The physical meaning of a "shallow" STG is not clear to the reader here. (I realize the description was taken out due to my request that a preceding sentence in the abstract be simplified.)

Thanks for pointing this out. We added the meaning of the 'shallow' STG in L20–22:

"A positive SIF anomaly coincides with a shallow STG, i.e., more positive (inversions) or less negative than the mean STG, since the temperature at the bottom of the valley decreases more than at the top."

L25-28: This sentence added to the revised abstract is confusing. Is the authors' hypothesis that the local conditions associated with anomalously low sea ice (i.e. a decrease in atmospheric stability and SMB) will become more prominent in the future with climate warming?

Thanks for the suggestion. We modified the sentence in L26–28:

"Based on our findings, we speculate that the local conditions in the Zackenberg region associated with anomalously low sea ice (i.e., a decrease in atmospheric stability) will be more prominent in the future with climate warming."

Be consistent with capitalization of the word "Arctic" throughout the manuscript. For example, it is not capitalized in L112 but is capitalized in L130.

Thanks for the suggestion. We now consistently capitalized the word Arctic throughout the manuscript.

L535-536: This sentence is a fragment. Do the authors mean that the potential relationships between these factors were examined?

Thanks for pointing this out. We added the missing fragment in the L435–436:

"In particular, the potential relationships between the STG, large-scale atmospheric circulation, surface and atmospheric moisture, and sea ice over the Greenland Sea were examined."

L1096-1097: I think it would be helpful to provide a little more detail about what the "significant implications" actually are. Are the authors making the point that their results show complex relationships between lower and higher elevations temperatures that vary with changes in local- and large-scale conditions, which should be considered by modeling studies that often only employ low elevation records?

Yes, and thanks for the suggestion. We added the following sentence at the beginning of the paragraph (L797–800):

"This study shows complex relationships between temperatures in low and high elevations that vary with changes in local- and large-scale conditions. Studies that calibrate vertical dependences of environmental variables based on low-elevation measurements and standard methods for spatial extrapolation may miss out on parts of the complexity and hence limit applicability."

**Technical corrections**

L15: "a" regional climate model

Done.

L16: "were" --> "was"

Done.

L16: "near fjord-ice condition" --> "near-fjord ice condition"

Done.

L22: "affect" --> "affects"

Done.

L92: "on" --> "in"

Done.

L94: "AWS" --> "AWSs"

Done.

L117: "anticyclone" --> "anticyclones"

Done.

L122: "warms" --> "warm"

Done.

L211: "exposition" --> "exposure"?

Done.

Figure 2 caption: "in" the ZR

Done.

L226: "was" --> "were"

Done.

L245: "data is" --> "data are"

Done.

L287: "predecessor" --> "predecessors"

Done.

L482: "exists" --> "exist"

Done.

L546: "are" --> "is"

Done.

L995: "condition" --> "conditions"

Done.